# TACKLING DIVERSE TASKS VIA CROSS-MODAL TRANSFER LEARNING

## ABSTRACT

Fine-tuning large-scale pretrained models has led to remarkable progress in well-studied modalities such as vision and NLP. However, similar gains have not been observed in many other tasks due to an *assumed* lack of relevant pretrained models for these diverse modalities. In this work, we revisit this assumption by studying the cross-modal transfer ability of large-scale pretrained models. We introduce ORCA, a general cross-modal fine-tuning workflow that enables fast and automatic exploitation of existing pretrained models for diverse tasks. ORCA achieves task-specific adaptation by performing data alignment before fine-tuning: it learns an embedding network that minimizes the optimal transport dataset distance between the end-task data and the pretraining data to close the modality gap. Through extensive experiments, we show that ORCA is the first viable approach that allows practitioners to use pretrained models to outperform hand-designed, AutoML-searched, and general-purpose architectures—ORCA obtains state-of-the-art results on 10 of 13 diverse tasks we evaluate and ranks among the top three on the others. We shed light on why cross-modal transfer works by quantifying the importance of data alignment and highlight ORCA's utility for data-limited domains.

## 1 INTRODUCTION

The success of machine learning (ML) in vision and natural language processing (NLP) has spurred its application beyond these traditional ML domains to diverse tasks such as solving partial differential equations (Li et al., 2021b), music modeling (Lewandowski et al., 2012), detecting cardiac disease (Hong et al., 2020), and many others. However, progress in these less-explored areas can be challenging due to (1) limited amounts of labeled data, (2) high computational cost and human effort for developing models from scratch, and (3) a lack of relevant large-scale pretrained models, which have in many cases obviated the first two issues in vision and NLP (e.g., Devlin et al., 2019; Carion et al., 2020; Dosovitskiy et al., 2021; Liu et al., 2021b; Radford et al., 2021).

There are two common approaches for practitioners to handle these issues: automated machine learning (AutoML) techniques (e.g., Roberts et al., 2021; Shen et al., 2022) that focus on designing task-specific networks in a data-efficient manner; and multimodal general-purpose methods that either propose flexible architectures applicable to various tasks (Jaegle et al., 2022a) or expand the set of modalities for which pretrained models exist (e.g., Reed et al., 2022; Lu et al., 2022a). However, both classes of approaches require training from scratch when applied to a new modality and proceed under the assumption of a lack of relevant pretrained models for these diverse problems.

In this work, we re-examine this assumption by considering the general problem of *cross-modal transfer*. Our goal is to exploit existing large-scale pretrained models in data-rich modalities for solving diverse downstream tasks. A few recent works have demonstrated the potential promise of cross-modal transfer by applying language transformers to vision (Kiela et al., 2019; Dinh et al., 2022; Lu et al., 2022b), referential games (Li et al., 2020c), and reinforcement learning (Reid et al., 2022). However, many of these approaches are ad-hoc (e.g., rely on manual prompt engineering or hand-craft new architecture components to solve specific tasks), and none of them yield models competitive with those trained from scratch. We tackle both shortcomings in our work.

We introduce a general-purpose, cross-modal transfer workflow called **ORCA** (**O**ptimal t**R**ansport **C**ross-modal **A**daptation) that yields state-of-the-art results on a wide range of non-text and non-vision problems using pretrained transformers (Figure 1). Our key insight is to align the feature

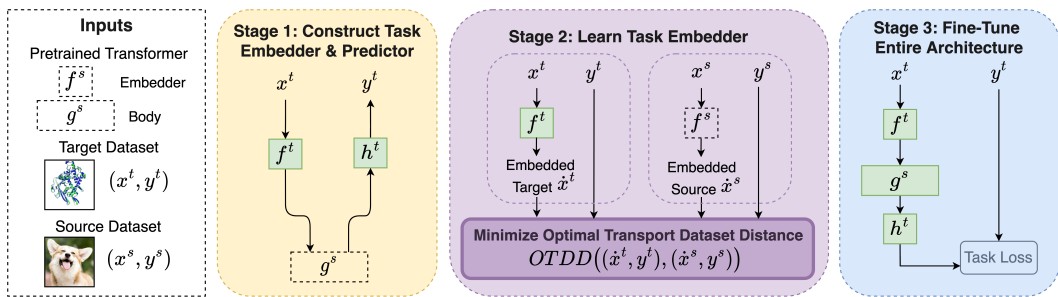

Figure 1: ORCA's three-stage cross-modal transfer workflow enables fast and automatic exploitation of large-scale pretrained models in data-rich modalities for solving diverse tasks. First, given target data $(x^t, y^t)$ and a pretrained transformer body $g^s$, ORCA constructs an embedder architecture $f^t$ to match the input dimensionality of $g_s$, and a predictor architecture $h^t$ to convert the output of $g^s$ back to the appropriate output space for the target task, e.g., classification logits or dense maps. Note that ORCA does *not* learn the weights for $f_t$ or $h_t$ during this stage. Next, ORCA learns the parameters for the embedder $f^t$ by minimizing the OTDD between the target dataset and an in-modality source dataset. Finally, ORCA fine-tunes the entire architecture $\{f^t, g^s, h^t\}$.

distribution of an unfamiliar, out-of-modality dataset with that of a familiar, in-modal dataset before fine-tuning. This data alignment process not only prevents distortion of pretrained weights but also enables cross-modal knowledge transfer, as we will show via extensive experiments in Section 4. Concretely, for any downstream task, we first generate an embedding network that maps the (potentially high-dimensional) inputs to sequence features. Then, we train it to minimize the optimal transport dataset distance (OTDD) (Alvarez-Melis & Fusi, 2020) between the feature-label distribution of the target data and data from the pretraining domain[1]. Finally, we fine-tune the pretrained model and the embedding network. Using OTDD allows us to relax many distributional assumptions required by traditional domain adaptation and perform data alignment using both the feature and label information of the target data. However, we show in an ablation study in Section 4.2.1 that substituting OTDD with other distance metrics, such as maximum mean discrepancy (MMD) (Gretton et al., 2012), can also aid cross-modal transfer, albeit to a lesser extent. This implies that it is the general idea of first-align-then-fine-tune that enables ORCA to obtain significantly better results than previous cross-modal learning methods that rely on vanilla fine-tuning (Lu et al., 2022b).

We evaluate ORCA on a diverse set of 13 tasks with different input dimensions (1D and 2D), prediction types (point and dense), and modalities (vision, audio, electrocardiogram, physics, protein, genomics, cosmic-ray, and music). ORCA outperforms various competitors, including task-specific hand-designed architectures, leading AutoML methods, and general-purpose models, ranking first on 10 tasks and in the top three on all tasks. We compare ORCA with existing fine-tuning techniques and confirm that effective cross-modal transfer is only enabled by ORCA's feature alignment process. We further reveal an empirical correlation between the alignment quality and the downstream performance. Finally, we demonstrate ORCA's efficacy for limited-data tasks. Overall, our work not only explores the cross-modal transfer ability of pretrained models, but also establishes a practical workflow for solving diverse prediction problems efficiently and automatically.

## 2 RELATED WORK

In this section, we review several groups of related work in the areas of *AutoML*, *in-modal transfer learning* (unimodal domain adaptation, unimodal/multimodal fine-tuning, and general purpose methods), and *cross-modal transfer learning* (heterogeneous domain adaptation, task-specific fine-tuning, and FPT). Table 1 summarizes these groups along relevant axes, and contrasts them to ORCA.

**AutoML for diverse tasks** is a growing research area, as evidenced by the NAS-Bench-360 benchmark (Tu et al., 2022), along with several recent neural architecture search (NAS) methods that target this problem, e.g., AutoML-Zero (Real et al., 2020), XD (Roberts et al., 2021), and DASH (Shen et al., 2022)). In contrast to these NAS methods, ORCA takes a transfer learning approach in order to leverage existing pretrained models from data-rich modalities for more esoteric tasks, rather than repeatedly incurring the overhead of designing new architectures and training them from scratch. That said, given the shared underlying motivation, our experimental evaluation makes use of the diverse

---

[1]We do not assume access to the pretraining data due to practical concerns about data access and computational efficiency. We instead work with publicly available proxy data from the pretraining modality, e.g., CIFAR-10 for models pretrained on ImageNet and CoNLL-2003 for models pretrained on larger text corpora.

Table 1: Summary of existing approaches aiming to develop models for diverse tasks.

| | | Task-specific adaptation? | General-purpose workflow? | Supports transfer to different: | | |
|---|---|:-:|:-:|:-:|:-:|:-:|
| | | | | input dim? | output dim? | modality? |
| Task-specific learning | Hand-designed models | ✓ | | | | |
| | AutoML models | ✓ | ✓ | | | |
| In-modal transfer | Unimodal DA | ✓ | | ✓ | | |
| | Uni/Multimodal fine-tuning | ✓ | | ✓ | ✓ | |
| | General-purpose models | ✓ | ✓ | ✓ | ✓ | |
| Cross-modal transfer | Heterogeneous DA | ✓ | | ✓ | | ✓ |
| | Task-specific fine-tuning | ✓ | | ✓ | ✓ | ✓ |
| | FPT | | ✓ | ✓ | ✓ | ✓ |
| | ORCA | ✓ | ✓ | ✓ | ✓ | ✓ |

tasks comprising NAS-Bench-360, and compares ORCA with its expert and AutoML baselines. We also compare against DASH, the state-of-the-art method on this benchmark.

**Unimodal domain adaptation (DA)** is a form of transductive transfer learning where the source and target tasks are the same but the domains differ (Pan & Yang, 2009; Wang & Deng, 2018). Many DA methods assume that the target data has the same input space and support as the source data, and are concerned with problems where the output spaces and the joint/marginal distributions differ, such as covariate and label shifts. Recent work considers more general settings such as different feature spaces (heterogeneous DA) or label spaces (universal DA). Our focus on cross-modal transfer goes one step further to the case where neither the input-space nor the output-space support overlaps.

**Unimodal fine-tuning** is a more flexible transfer approach that can be applied to downstream tasks with different label spaces or input spaces. Pretrained models are used for in-modality fine-tuning in NLP (e.g., Aghajanyan et al., 2021; Jiang et al., 2020), vision (e.g., Wei et al., 2022; Li et al., 2022), speech (e.g., Chen et al., 2022; Jiang et al., 2021), protein sequences (Jumper et al., 2021), and robotics (Ahn et al., 2022). Adapter networks (He et al., 2022) have been developed to improve the downstream performance of in-modality transfer. **Multimodal fine-tuning** expands the applicable modalities of a single pretrained model by learning embeddings of several data-rich modalities together (e.g., Lu et al., 2019; Radford et al., 2021; Hu & Singh, 2021; Kim et al., 2021; Alayrac et al., 2022). However, these approaches still focus on solving in-modality downstream tasks.

**General-purpose models** propose flexible architectures applicable to various tasks such as optical flow, point clouds, and reinforcement learning (Jaegle et al., 2021; 2022a; Reed et al., 2022). These approaches train multitask transformers from scratch using a large body of data from different tasks. Though more versatile than unimodal models, they still focus on transferring to problems within the pretraining modalities considered. Nonetheless, the success of transformers for in-modality fine-tuning motivates us to focus on adapting transformer-type architectures for cross-modal transfer.

**Heterogeneous DA (HDA)** considers nonequivalent feature spaces between the source and target domains. While most HDA methods are developed for same-modality-different-dimension transfer, e.g., between images of different resolutions, there are indeed a few works studying cross-modal tasks such as text-to-image (Yao et al., 2019; Li et al., 2020b). However, a crucial assumption that HDA makes is that the target and source tasks are the same. Thus, we operate in a much more flexible setting and consider knowledge transfer between drastically different domains with distinct tasks and label sets, such as applying Swin Transformers (Liu et al., 2021c) to solving partial differential equations or RoBERTa to classifying satellite images and electrocardiograms.

**Cross-modal, task-specific fine-tuning** is a recent line of research, with most work focusing on transferring NLP models to other modalities like vision (Kiela et al., 2019), referential games (Li et al., 2020c), and reinforcement learning (Reid et al., 2022). These works provide initial evidence of the cross-modal transfer capacity of pretrained models. However, they focus on hand-tailoring to a single modality, e.g., by adding ad-hoc encoders that transform agent messages (Li et al., 2020c) or decision trajectories (Reid et al., 2022) into tokens. Even when not relying on fine-tuning, work like LIFT (Dinh et al., 2022) that attempts cross-modal learning via prompting (Liu et al., 2021a) still require ad-hoc conversion of tasks to natural text.

**Frozen Pretrained Transformers (FPT)** (Lu et al., 2022b) is a general cross-modal fine-tuning workflow that transforms input features to be compatible with the pretrained models. Although FPT and ORCA are both general-purpose workflows, FPT does not account for differences between the target and pretraining modalities, which we show is necessary to achieve accurate predictive models and outperform existing baselines.

## 3   ORCA WORKFLOW

In this section, we first formalize the problem setup and then introduce the ORCA workflow for adapting pretrained transformers to diverse end tasks.

**Problem Setup.** A domain $\mathcal{D}$ consists of a feature space $\mathcal{X}$, a label space $\mathcal{Y}$, and a joint probability distribution $P(\mathcal{X}, \mathcal{Y})$. In the cross-modal setting we study, the target (end-task) domain $\mathcal{D}^t$ and source (pretraining) domain $\mathcal{D}^s$ differ not only in the feature space but also the label space and by extension have differing probability distributions, i.e., $\mathcal{X}^t \neq \mathcal{X}^s$, $\mathcal{Y}^t \neq \mathcal{Y}^s$, and $P^t(\mathcal{X}^t, \mathcal{Y}^t) \neq P^s(\mathcal{X}^s, \mathcal{Y}^s)$. This is in contrast to the transductive transfer learning setting addressed by domain adaptation, where source and target domains share the label space and end task (Pan & Yang, 2009).

Given target data $\{x_i^t, y_i^t\}_{i \in [n^t]}$ sampled from a joint distribution $P^t$ in domain $\mathcal{D}^t$, our goal is to learn a model $m^t$ that correctly maps each input $x^t$ to its label $y^t$. We are interested in achieving this using pretrained transformers. Thus, we assume access to a model $m^s$ that has been trained with data $\{x_i^s, y_i^s\}_{i \in [n^s]}$ in the source domain $\mathcal{D}^s$, where $(x_i^s, y_i^s) \sim P^s$. Then, given a predefined loss function $l$, we aim to develop $m^t$ based on $m^s$ such that $L(m^t) = \mathbb{E}_{(x^t, y^t) \sim P^t}[l(m^t(x^t), y^t)]$ is minimized. This problem formulation does not define modality explicitly and includes both in-modal and cross-modal transfer. Given the generality of the tasks we wish to explore, it is hard to provide a precise mathematical definition, so we rely on semantics to differentiate the two settings: intuitively, cross-modal domains (e.g., natural images vs. protein sequences) are more distinct to each other than in-modal domains (e.g., photos taken in two different geographical locations).

Having defined the learning problem, we now present our three-stage cross-modal transfer workflow: (1) architecture design to support diverse input-output dimensions, (2) embedder pretraining to align the source and target feature distributions, and (3) fine-tuning to minimize the target task loss.

### 3.1   TASK-SPECIFIC ARCHITECTURE DESIGN

Applying pretrained models to another downstream problem usually requires addressing the problem of mismatched dimensions. To make ORCA work for input and output tensors of different dimensions, we decompose a transformer-based learner $m$ into three parts (Figure 1 stage 1): an embedder $f$ that transforms input $x$ into a sequence of features, a model body $g$ that applies a pretrained transformer (i.e., series of attention layers) to the embedded features, and a predictor $h$ that generates predictions with the desired output shape. ORCA uses pretrained architecture and weights to initialize the model body $g$ but replaces $f$ and $h$ with layers designed to match the target data with the pretrained model's embedding dimension. Next, we describe each module in detail.

**Custom Embedding Network.** Denote the feature space compatible with the pretrained model body as $\dot{\mathcal{X}}$. For a transformer with maximum sequence length $S$ and embedding dimension $D$, $\dot{\mathcal{X}} = \mathbb{R}^{S \times D}$. The embedding network $f : \mathcal{X} \to \dot{\mathcal{X}}$ is designed to take in a tensor of arbitrary dimension from $\mathcal{X}$ and transform it to the feature space $\dot{\mathcal{X}}$. In ORCA, $f$ is composed of a convolutional layer with input channel $c_{in}$, output channel $c_{out}$, kernel size $k$, and stride $k$, generalizing the patching operations used in vision transformers to 1D and higher-dimensional cases. We set $c_{in}$ to the input channel of $x$ and $c_{out}$ to the embedding dimension $D$. To take full advantage of the representation power of the pretrained model, we choose the smallest $k$ for which the product of output shape excluding the channel dimension $\leq S$. That is, when we flatten the non-channel dimensions of the output tensors after the convolution, pad and then transpose it, we can obtain sequence features with shape $S \times D$. Finally, we add a layer norm and a positional embedding to obtain $\dot{x}$.[2]

**Pretrained Transformer Body.** The model body $g$ takes the embedding $\dot{x} \in \dot{\mathcal{X}}$ as input and outputs features $\dot{y} \in \dot{\mathcal{Y}}$; the dot is used to differentiate these intermediate representations from the raw inputs and labels. For transformer-based $g$, both the input and output feature spaces $\dot{\mathcal{X}}, \dot{\mathcal{Y}}$ are $\mathbb{R}^{S \times D}$.

**Custom Prediction Head.** Finally, the prediction head $h$ must take $\dot{y} \in \dot{\mathcal{Y}}$ as input and return a task-dependent output tensor. Different tasks often specify different types of outputs, e.g., classification tasks require logits in $\mathbb{R}^K$ where $K$ is the number of classes, and dense prediction tasks require dense maps with the same spatial dimension as the input and per index logits correspond-

---

[2]As a concrete example, consider an image tensor with shape $(C_{in}, H_{in}, W_{in})$. We first choose stride $k$ for the convolution such that $H_{out} \times W_{out} \approx S$ to get an output tensor with shape $(D, H_{out}, W_{out})$. Then, we flatten it to shape $(D, H_{out} \times W_{out})$, pad along the last dimension to shape $(D, S)$, and transpose.

ing to $K$ classes. Thus, it is crucial to define task-specific output modules and fine-tune them when transferring to new tasks. In our workflow, we use the simplest possible instantiation of the predictor modules. For classification, we apply average pooling along the sequence length dimension (or take the classification token of language models) to obtain 1D tensors with length $D$ and then use a linear layer that maps $D$ to $K$. For dense prediction, we apply a linear layer to the sequence outputs so the resulting tensor has shape $(S, k^{\dim(\mathcal{Y})-1}K)$, where $k^{\dim(\mathcal{Y})-1}$ is the downsampling factor of the embedder convolution kernel with stride $k$. This upsamples by the same factor that the embedder convolution downsampled. Then, we can mold the tensor to the desired output dimension.[3]

With an architecture based on the pretrained model but is also compatible with our target task, we can now turn our attention to pretraining the embedder via matching source and target distributions.

## 3.2 Embedding Learning dor Data Alignment

Intuitively, transferring knowledge across similar modalities should be easier than across distant ones. Hence, given a target task in a new modality, we aim to manipulate the task data so that they become closer to the pretraining modality. We use the optimal transport dataset distance (OTDD) (Alvarez-Melis & Fusi, 2020) to measure the closeness between datasets in different domains. Unlike the classic OT distance which operates only on the *feature* distributions, OTDD considers both the feature and the *label* information and can work even if *the label sets are unrelated or disjoint*. Thus, while OT is mostly used for unsupervised or semi-supervised domain adaptation (Courty et al., 2017; Yan et al., 2018), OTDD is particularly suitable for distance estimation between cross-modal labeled datasets. In the following, we will briefly explain how ORCA uses OTDD. We refer the readers to Alvarez-Melis & Fusi (2020) for a detailed exposition on the metric itself.

Formally, let $f^s : \mathcal{X}^s \to \dot{\mathcal{X}}$ denote the pretrained embedder (the part of $m^s$ that transforms the source data to sequence features) and $f^t : \mathcal{X}^t \to \dot{\mathcal{X}}$ be a randomly initialized target embedder with architecture discussed in the previous section. We train $f^t$ to minimize the expected OTDD between the embedding-label distributions $\big(f^t(x^t), y^t\big)$ and $\big(f^s(x^s), y^s\big)$. That is, for both datasets, we first represent each class label as a distribution over the in-class features: $y \mapsto P(\dot{\mathcal{X}}|\mathcal{Y} = y)$. This transforms the source and target label sets into the shared space of distributions over $\dot{\mathcal{X}}$. Then, we can define the distance $d_{\mathcal{Y}}(y^t, y^s)$ between different labels using the $p$-Wasserstein distance associated with a metric $d_{\dot{\mathcal{X}}}$ over the feature space, e.g., the $l_2$ distance $\|\dot{x}^t - \dot{x}^s\|_2^2$. This allows us to measure the difference between distributions in $\dot{\mathcal{X}} \times \mathcal{Y}$ using the following $p$-Wasserstein metric:

$$d_{\dot{\mathcal{X}} \times \mathcal{Y}}\big((\dot{x}^t, y^t), (\dot{x}^s, y^s)\big) = \big(d_{\dot{\mathcal{X}}}(\dot{x}^t, \dot{x}^s)^p + d_{\mathcal{Y}}(y^t, y^s)^p\big)^{1/p}. \tag{1}$$

Plugging this into the OT formulation leads to the OTDD over $\dot{\mathcal{X}} \times \mathcal{Y}$, which we optimize to learn $f^t$.

Leveraging the clustering structure of the datasets, OTDD provides a better distance estimation between the target and source data, demonstrating better alignment ability in practice. In Section 4.2.1, we examine several distance metrics for learning the embedder. While data alignment generally improves downstream performance for all metrics, OTDD leads to the best empirical results.

As for the computational cost of embedding learning, we analyze the complexity of OTDD in Appendix A.1 and show that this stage takes much less time than the later fine-tuning stage in Appendix A.5. Thus, ORCA achieves a significant performance gain at a low cost for feature alignment.

## 3.3 Fine-Tuning for Downstream Adaptation

After training the embedder, we perform full fine-tuning by updating all model parameters to minimize the target loss. This step further aligns the embedder and predictor with the pretrained model to improve downstream performance. We perform an ablation study comparing ORCA to standard fine-tuning without feature matching in Section 4.2.1 and show that our approach improves prediction accuracy and reduces performance variance. There are orthogonal lines of work that study how to best fine-tune a pretrained model (e.g., Liu et al., 2022; He et al., 2022). We compare with one strategy used in FPT (Lu et al., 2022b) in Section 4.2.2 but leave further exploration for future work.

---

[3]As a concrete example, for an image tensor with embedding convolution kernel size $k$, the linear layer will yield an output of shape $(S, k^2K)$, which we transpose, pad, and reshape to $(k^2K, H_{out}, W_{out})$. Finally, we apply pixelshuffle (Shi et al., 2016) to get an output of shape $(K, H_{in}, W_{in})$.

## 4 EXPERIMENTS

Having introduced how ORCA addresses the dimension mismatch between the target and source datasets via architecture design and tackles the distribution mismatch via embedder learning, we now proceed with showing its empirical effectiveness. In the following, we will demonstrate that ORCA is the first approach that allows practitioners to obtain models better than hand-designed, AutoML-searched, and general-purpose architectures on a variety of diverse tasks. Then, we will analyze key components of ORCA to better understand the mechanism underlying cross-modal transfer.

**Experiment Protocol.** While our workflow accepts a wide range of pretrained transformers as model bodies, we use RoBERTa (Liu et al., 2019b) and Swin Transformers (Liu et al., 2021c), which are representatives of the most studied language and vision modalities, to exemplify ORCA's efficacy. We implement the base models, which have around 100 million parameters, and use the pretrained weights available in the Hugging Face transformers library (Wolf et al., 2019). As stated in the introduction, we do not use the exact pretraining data to represent the source modalities because they are often not publicly available and can be too large to compute OTDD efficiently. We use the proxy datasets: CoNLL-2003[4] for RoBERTa and CIFAR-10 for Swin.

For each task, we first apply the hyperparameter tuning algorithm ASHA (Li et al., 2020a) to the standard fine-tuning baseline ("Fine-tuning" in Table 3) to identify suitable batch size, optimizer, learning rate, and weight decay. These hyperparameters are then applied to all fine-tuning baselines as well as ORCA. During embedder learning, while classification tasks naturally come with discrete labels required for computing OTDD, for dense prediction tasks where labels are high-dimensional maps, we perform clustering on the dense maps to generate pseudo labels, which not only preserves the intrinsic distribution of the target data but also speeds up OTDD computation. We manage our experiments using the Determined AI platform. All experiments are performed on NVIDIA V100 GPUs and results are averaged over 5 random seeds. For other experiment details, see Appendix A.3.

### 4.1 CAN PRETRAINED MODELS TRANSFER ACROSS MODALITY TO SOLVE DIVERSE TASKS?

In this section, we highlight the most important observation of this work: cross-modal fine-tuning with ORCA can solve a variety of tasks effectively and efficiently. To demonstrate this, we evaluate ORCA on 13 tasks detailed below. We first include 10 tasks from NAS-Bench-360[5], which covers problems such as PDE solving, protein folding, and cardiac disease detection. This benchmark contains tasks for 1D and 2D classification, 2D dense prediction, but not 1D dense prediction, so we added JSB Chorales, a music modeling dataset widely used for evaluating recurrent networks (Chung et al., 2017; Bai et al., 2018). We also added ListOps (parsing math expressions) (Tay et al., 2021) and Homology (classifying protein structure) (Rao et al., 2019) for comparison with FPT. Together, these 13 tasks represent a wide collection of modalities for comprehensive evaluation.

Following the taxonomy in Table 1, we consider three classes of baselines: (1) hand-designed expert architectures for each task, as identified by Tu et al. (2022), Rao et al. (2019), and Tay et al. (2021); (2) general-purpose models, as represented by Perceiver IO (Jaegle et al., 2022b); and (3) AutoML baselines, as represented by those evaluated in NAS-Bench-360 and DASH (Shen et al., 2022). We will compare with FPT later, the only remaining approach with a general workflow from Table 1.

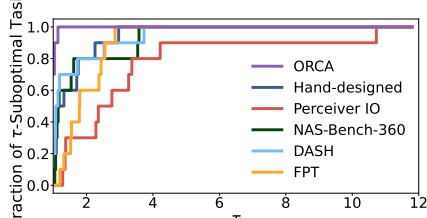

In Table 2, we report the prediction error for each method on each task. ORCA achieves the lowest error rate on 10 of 13 tasks and is the most effective in terms of aggregated performance. This is also supported by the performance summary in Figure 2. More specifically, we outperform all hand-designed architectures on all tasks except ECG, where we rank second but do much better than the other

Figure 2: Aggregate performance of ORCA and baselines for Table 2 results measured by performance profiles (Dolan & Moré, 2002). Larger values (larger fractions of tasks on which a method is within $\tau$-factor of the best) are better. ORCA being in the top left corner means it is often the best.

---

[4]CoNLL-2003 is for named entity recognition. It is used to interpret language models (Jawahar et al., 2019).

[5]NAS-Bench-360 is designed for testing how well ML algorithms can generalize and is a core component of the 2022 AutoML Decathlon competition. For a summary of included tasks, see Table 6 in the Appendix

Table 2: Prediction errors (lower is better) for 13 tasks across diverse application domains. On 10/13 problems, ORCA ranks the first among all hand-designed expert models, AutoML and general-purpose methods. NAS-Bench-360 refers to the task-wise best of all baselines evaluated in the benchmark, including DARTS (Liu et al., 2019a), DenseNAS (Fang et al., 2020), XGBoost (Chen & Guestrin, 2016), and 3 others. FPT refers to fine-tuning the layer norms of RoBERTa or Swin. For a list of hand-designed models, see Appendix A.2.

| | CIFAR-100 0-1 error (%) | Spherical 0-1 error (%) | Darcy Flow relative $\ell_2$ | PSICOV $MAE_8$ | Cosmic 1-AUROC | NinaPro 0-1 error (%) | FSD50K 1- mAP |
|---|---|---|---|---|---|---|---|
| Hand-designed | 19.39±0.20 | 67.41±0.76 | 8E-3±1E-3 | 3.35±0.14 | 0.24±0.015 | 8.73±0.9 | 0.62±0.004 |
| NAS-Bench-360 | 23.39±0.01 | 48.23±2.87 | 2.6E-2±1E-3 | 2.94±0.13 | 0.22±0.035 | 7.34±0.76 | 0.60±0.001 |
| DASH | 24.37±0.81 | 75.44±2.38 | 7.9E-3±2E-3 | 3.30±0.16 | 0.25±0.02 | **6.60±0.33** | 0.60±0.008 |
| Perceiver IO | 70.04±0.44 | 82.57±0.19 | 2.4E-2±1E-2 | 8.06±0.06 | 0.48±0.01 | 22.22±1.80 | 0.72±0.002 |
| FPT-Swin | 10.11±1.18 | 76.38±4.89 | 2.1E-2±1.3E-3 | 4.66±0.054 | 0.32±0.0021 | 15.69±2.33 | 0.67±0.0068 |
| **ORCA-SWIN** | **6.53±0.079** | **29.85±0.72** | **7.3E-3±6.8E-5** | **1.91±0.038** | **0.21±0.0050** | 7.54±0.39 | **0.56±0.013** |
| | ECG 1 - F1 score | Satellite 0-1 error (%) | DeepSEA 1- AUROC | JSB Chorales NLL | ListOps 0-1 error (%) | Homology 0-1 error (%) | |
| Hand-designed | **0.28±0.00** | 19.8±0.00 | 0.30±0.024 | 8.43±0.00 | 62.73±0.00 | 88±0.00 | |
| NAS-Bench-360 | 0.33±0.02 | 12.51±0.24 | 0.32±0.01 | - | - | - | |
| DASH | 0.32±0.007 | 12.28±0.5 | **0.28±0.013** | 6.13±0.006 | 57.33±0.14 | 89.71±0.54 | |
| Perceiver IO | 0.66±0.01 | 15.93±0.008 | 0.38±0.004 | - | - | - | |
| FPT-RoBERTa | 0.50±0.0098 | 20.83±0.24 | 0.37±0.0002 | 2.72±0.026 | 64.35±0.79 | 91.48±1.14 | |
| **ORCA-ROBERTA** | 0.29±0.0052 | **11.59±0.18** | 0.29±0.006 | **2.44±0.056** | **51.90±2.18** | **87.21±0.50** | |

methods. We also beat all AutoML baselines on all tasks except DeepSEA and NinaPro, where ORCA is second and third, respectively. The improvements from ORCA come at a small computational overhead associated with pretraining the embedder to match the source and target modalities. Table 5 in the Appendix shows the time needed for embedder learning with OTDD, which is a small portion (10.2% on average) of the fine-tuning time. ORCA's efficiency and its state-of-the-art results on 10 tasks make it a practical tool for model development in diverse areas.

Our experiments further validate the findings in Lu et al. (2021) that pretrained transformers can learn knowledge transferable to seemingly unrelated tasks. In the following, we delve into the mechanism of ORCA to provide intuition for necessary components of successful cross-modal learning.

## 4.2 KEY FACTORS FOR SUCCESSFUL CROSS-MODAL TRANSFER

Here, we dissect the success of cross-modal transfer with ORCA through a series of ablation studies. As a preview, we identify three aspects to be key to its success: data alignment through embedding learning, full fine-tuning of all model weights, and suitable pretrained model selection.

### 4.2.1 MATCHING FEATURE DISTRIBUTIONS HELPS ADAPTATION

Table 2 shows that ORCA leads to effective transfer using the proposed three-stage workflow. However, as learning the embedder via OTDD is an instantiation of the general first-align-then-fine-tune paradigm, we ask the question: *does cross-modal transfer work because of the specific application of OTDD or the core approach of data alignment across modalities?*

To answer this question, we perform an ablation study on the embedding learning metrics and compare their performance to fine-tuning without embedder pretraining. We experiment with OTDD, maximum mean discrepancy (MMD) (Gretton et al., 2012), the mutual-information-based TransRate (Huang et al., 2022), and Euclidean distance. The performance profile is in shown Figure 3, and the detailed results are shown in Appendix A.4. We highlight the following observations. First, bringing the target modality closer to the pretraining modality generally aids cross-modal transfer, regardless of which metric we minimize. This is evident from the fact that pretraining the embedder with any of the metrics can outperform vanilla fine-tuning without embedder learning on many

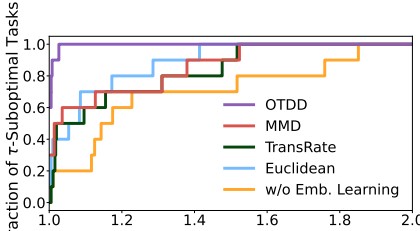

Figure 3: Performance profiles for using different embedder learning metrics on NAS-Bench-360. Data alignment generally improves fine-tuning performance.

tasks. Second, among the evaluated metrics, OTDD leads to the best overall performance. This is why we use it in our workflow. The middle rows of Table 3 demonstrate that ORCA with OTDD consistently outperforms naive fine-tuning. This supports our argument that closing the gap between a new modality and the pretraining modality can facilitate a model's adaptation to a new task.

Table 3: Performance of ORCA, vanilla fine-tuning, and training RoBERTa/Swin from scratch. We consider fine-tuning all parameters (full setting) vs. only the layer norms (FPT setting). ORCA is better in both settings.

| | CIFAR-100 | Spherical | Darcy Flow | PSICOV | Cosmic | NinaPro | FSD50K |
|---|---|---|---|---|---|---|---|
| Train-from-scratch | 50.87±0.32 | 76.67±0.21 | 8.0E-2±1.3E-2 | 5.09±0.014 | 0.50±0.00 | 9.96±1.67 | 0.75±0.017 |
| Fine-tuning | 7.67±0.55 | 55.26±1.63 | **7.3E-3±1.1E-4** | 1.92±0.039 | 0.24±0.0080 | 8.35±0.75 | 0.63±0.014 |
| **ORCA** | **6.53±0.079** | **29.85±0.72** | **7.3E-3±6.8E-5** | **1.91±0.038** | **0.21±0.0050** | **7.54±0.39** | **0.56±0.013** |
| Fine-tuning (layernorm) | 10.11±1.18 | 76.38±4.89 | 2.1E-2±1.3E-3 | 4.66±0.054 | 0.32±0.0021 | 15.69±2.33 | 0.67±0.0068 |
| **ORCA (layernorm)** | 7.99±0.098 | 42.45±0.21 | 2.1E-2±7.4E-4 | 4.97±0.14 | 0.25±0.0020 | 15.99±1.92 | 0.64±0.0093 |

| | ECG | Satellite | DeepSEA | JSB Chorales | ListOps | Homology | |
|---|---|---|---|---|---|---|---|
| Train-from-scratch | 0.42±0.011 | 12.38±0.14 | 0.39±0.01 | 2.71±0.35 | 63.50±0.55 | 90.39±0.13 | |
| Fine-tuning | 0.44±0.0056 | 13.86±1.47 | 0.51±0.0001 | 2.59±0.098 | 72.79±1.96 | 89.31±0.60 | |
| **ORCA** | **0.29±0.0052** | **11.59±0.18** | **0.29±0.006** | **2.44±0.056** | **51.90±2.18** | **87.21±0.50** | |
| Fine-tuning (layernorm) | 0.50±0.0098 | 20.83±0.24 | 0.37±0.0002 | 2.72±0.026 | 64.35±0.79 | 91.48±1.14 | |
| **ORCA (layernorm)** | 0.47±0.007 | 20.54±0.49 | 0.36±0.0070 | 2.68±0.13 | 63.62±0.75 | 90.65±1.66 | |

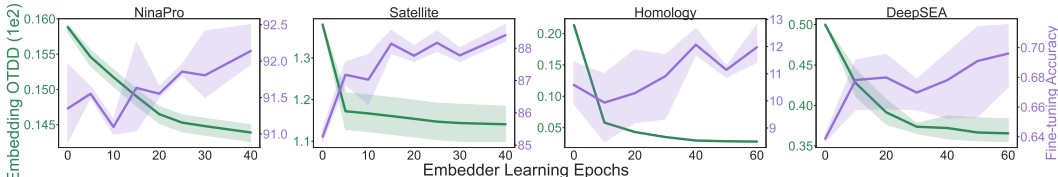

Figure 4: Final model accuracy and embedding OTDD vs. number of embedding training epochs when applying ORCA to four tasks. As the embedder learns to map the target data to the source modality better (smaller final OTDD), we generally obtain models with better downstream performance.

To further isolate the impact of data alignment, we compare ORCA with a train-from-scratch baseline, which trains RoBERTa and Swin using only the target data for the same number of epochs. Table 3 shows that train-from-scratch is better than fine-tuning but worse than ORCA on many tasks like Satellite and DeepSEA, indicating that when the target modality differs significantly from the pretraining modality, *naive fine-tuning may harm transfer*, but aligning the feature distribution using ORCA can resolve this issue and benefit transfer. Indeed, recent work has shown that optimizing directly for the task loss may distort the pretrained weights and lead to suboptimal solutions (Kumar et al., 2022; Lee et al., 2022). By manipulating the target data distribution to look like the source distribution, we can lower the risk of catastrophic forgetting, which may explain the success of ORCA.

Lastly, we perform experiments to quantify the effect of data alignment from a task-wise perspective. We train the embedder for different number of epochs before fine-tuning to see how optimizing OTDD to various levels of convergence affects downstream performance. Figure 4 plots the fine-tuning accuracy along with the final OTDD objective for different levels of embedder pretraining. Evidently, as the dataset distance decreases, the final fine-tuning accuracy increases. This correlation supports the effectiveness of embedder learning for cross-modal transfer. In addition, we observe that learning the embedder prior to fine-tuning can stabilize training, as the performance variance of ORCA is consistently lower than that of standard fine-tuning.

### 4.2.2 FINE-TUNING ALL MODEL PARAMETERS FOR CROSS-MODAL TASKS

As discussed in Section 2, Frozen Pretrained Transformers (FPT) (Lu et al., 2022b) is a related work that showed pretrained language models contain knowledge relevant for out-of-modality tasks. While FPT presented a general pipeline that transfers GPT-2 to tasks like CIFAR-10, Homology, and ListOps, the resulting models were not as good as those directly trained on the target data. FPT differs from ORCA in that (1) it does not pretrain an embedder for task-specific adaptation and (2) it only fine-tunes the layer norms. We have already verified the importance of (1). Now, to isolate the impact of (2), we evaluate ORCA with fine-tuning the layer norms vs. FPT on our task set.

The bottom rows of Table 3 show ORCA with fine-tuning just the layer norms outperforms FPT, indicating pretraining the embedding layers boosts the cross-modal performance of FPT. However, this performance gain is smaller than that seen in the full fine-tuning setting, which implies that full fine-tuning can take better advantage of the learned embeddings. Also, partial fine-tuning is less effective than full fine-tuning on all tasks except for DeepSEA. This exception might be due to the fact that full fine-tuning without learned embeddings is more prone to overfitting. In terms of runtime, FPT only results in less than 2x speedups compared with full fine-tuning (see Appendix A.5), despite the fact that we are updating significantly fewer parameters. This is unsurprising since gradients are still back-propagated through the entire network. Therefore, when computation allows, we recommend using ORCA with full fine-tuning for better downstream performance.

### 4.2.3 PRETRAINING MODALITY CAN AFFECT TRANSFER PERFORMANCE

Finally, we study how the pretraining modality affects fine-tuning performance. For experiments in Table 2, we chose pretrained models for each task based on the input dimension, i.e., we use RoBERTa for all 1D tasks and Swin for all 2D tasks. Now, we can switch the model bodies and apply ORCA. This is easy to implement because ORCA is model-agnostic and the embedder architec-

Table 4: Test errors of ORCA on DeepSEA and Spherical with language- and image-pretrained model bodies. Numbers in the parenthesis represent the OTDD after embedding learning. Smaller OTDD leads to better performance.

| Error (OTDD) | DeepSEA (1D) | Spherical (2D) |
|---|---|---|
| RoBERTa (1D) | **0.295±0.006 (37.40)** | 68.28±0.017 (19.54) |
| Swin (2D) | 0.361±0.001 (64.83) | **29.85±0.072(11.78)** |

ture handles all necessary input transformation to obtain sequence features. As shown in Table 4, fine-tuned RoBERTa outperforms fine-tuned Swin on the 1D task, and the final OTDD objective for RoBERTa is also smaller than that of Swin. We hypothesize that this is because the considered DeepSEA data (genomics sequences) are structured more like language than images with discrete units of information and general grammatical rules. The FPT paper observes a similar trend for Homology. As for the 2D tasks, we again notice that models with better fine-tuning accuracy have smaller OTDDs. This suggests a way of selecting pretrained models from a predefined model hub for each task, e.g., by comparing the optimized OTDDs and picking the one with the smallest value.

**Case Study: Low-Data Regime.** Now that we have a better understanding of ORCA, recall that one of our motivations for transferring pretrained models to various modalities is to help task-solving in data-limited regimes, where training models from scratch can be challenging. To this end, we investigate whether ORCA can facilitate fine-tuning large-scale models on small target datasets.

Indeed, for vanilla fine-tuning, a small amount of data may not give enough signal to update the pretrained weights. However, it is possible to obtain a good feature embedder with the same amount of data using ORCA, which can then make fine-tuning easier. In Figure 5, we vary the amount of target data and plot the performance of ORCA and vanilla fine-tuning. The performance gain of ORCA increases as the amount of data used decreases. This shows that fine-tuning does suffer from limited data, but ORCA can considerably alleviate the problem and improve downstream performance. Moreover, ORCA allows us to use a third of the data to match the performance of standard fine-tuning. Thus, it can benefit model development in domains where data collection is costly.

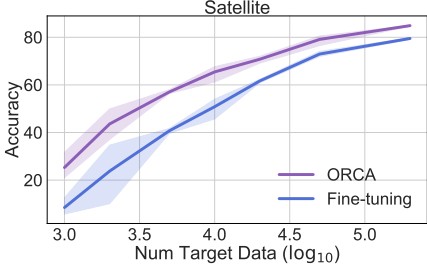

Figure 5: Prediction accuracy of ORCA and the fine-tuning baseline trained with different amounts of target data. ORCA has a larger performance gain in low-data regime.

**Discussion and Future Work.** We identify several future directions based on our experiment results. First, it is worth studying the effect of pretraining modality further and develop a systematic way of selecting pretrained models. Then, we can incorporate model selection into ORCA for a more automated transfer pipeline. Second, while ORCA leverages the simplest fine-tuning paradigm, we believe it is possible to combine it with more sophisticated transfer techniques such as adapters (He et al., 2022). We briefly study how prompting (Bahng et al., 2022; Jia et al., 2022) can be applied to diverse tasks in Appendix A.6 and find that it is in general less effective for out-of-modality problems, so we can possibly boost its performance using ORCA. Lastly, we currently evaluate ORCA on diverse 1D/2D tasks and in-modality vision tasks (Appendix A.7). It is also important to validate it on more settings, such as high-dimensional problems and reinforcement learning (Reid et al., 2022).

## 5 CONCLUSION

In this paper, we argue that an important step towards developing more general ML methods is to study how we can reuse existing models effectively for new and less-explored tasks. To this end, we propose a novel framework that allows transferring pretrained transformers to distinct downstream modalities. Our method, ORCA, can map target data from an arbitrary end task's modality to a model's pretraining modality to improve fine-tuning performance. We believe that this work not only signals the potential of large-scale pretraining for diverse tasks but also lays out a path for a largely uncharted data-centric paradigm in machine learning.

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

# A    APPENDIX

## A.1    EMBEDDING LEARNING WITH OPTIMAL TRANSPORT DATASET DISTANCE

### A.1.1    LITERATURE REVIEW

Due to the limited space, we do not give a full review of the optimal transport dataset distance (OTDD) (Alvarez-Melis & Fusi, 2020) in the main text. Here, we briefly recall the optimal transport (OT) distance and explain OTDD in detail.

Consider a complete and separable metric space $\mathcal{X}$ and let $\mathcal{P}(\mathcal{X})$ be the set of probability measures on $\mathcal{X}$. For $\alpha, \beta \in \mathcal{P}(\mathcal{X})$, let $\Pi(\alpha, \beta)$ be the set of joint probability distributions on $\mathcal{X} \times \mathcal{X}$ with marginals $\alpha$ and $\beta$ in the first and second dimensions respectively. Then given a cost function $c(\cdot, \cdot) : \mathcal{X} \times \mathcal{X} \to \mathbb{R}^{+}$, the classic OT distance with cost $c$ is defined by:

$$\text{OT}_c(\alpha, \beta) := \min_{\pi \in \Pi(\alpha, \beta)} \int_{\mathcal{X} \times \mathcal{X}} c(x, y) d\pi(x, y). \tag{2}$$

When $\mathcal{X}$ is equipped with a metric $d_{\mathcal{X}}$, we can use $c(x, y) = d_X(x, y)^p$ for some $p \geq 1$ and obtain the $p$-Wasserstein distance, $W_p(\alpha, \beta) := (\text{OT}_{d_{\mathcal{X}}^p}(\alpha, \beta))^{\frac{1}{p}}$.

Now consider the case of finite datasets with features in $\mathcal{X}$ and labels in a finite set $\mathcal{Y}$. Each dataset can be considered a discrete distribution in $\mathcal{P}(\mathcal{X} \times \mathcal{Y})$. To define a distance between datasets, a natural approach is to define an appropriate cost function on $\mathcal{Z} := \mathcal{X} \times \mathcal{Y}$ and consider the optimal transport distance. Indeed, for any metric $d_{\mathcal{Y}}$ on $\mathcal{Y}$ and any $p \geq 1$, $\mathcal{Z}$ can be made a complete and separable metric space with metric

$$d_{\mathcal{Z}}((x, y), (x', y')) = (d_{\mathcal{X}}(x, x')^p + d_{\mathcal{Y}}(y, y')^p)^{\frac{1}{p}} \tag{3}$$

It is usually not clear how to define a natural distance metric in $\mathcal{Y}$, so instead we proceed by representing each class $y \in \mathcal{Y}$ by $P(\mathcal{X}|\mathcal{Y} = y)$, the conditional distribution of features $\mathcal{X}$ given $\mathcal{Y} = y$. More specifically, for a dataset $\mathcal{D} \in \mathcal{P}(\mathcal{X} \times \mathcal{Y})$, denote this map from classes to conditional distributions by $F(\mathcal{D}, \cdot) : \mathcal{Y} \to \mathcal{P}(\mathcal{X})$. Then we can transform any dataset over $\mathcal{X} \times \mathcal{Y}$ into one over $\mathcal{X} \times \mathcal{P}(\mathcal{X})$ via $G(\mathcal{D}) := (\text{proj}_{\mathcal{X}}, F(\mathcal{D}, \text{proj}_Y))$.

As discussed above, $W_p$ is a natural notion of distance in $\mathcal{P}(\mathcal{X})$, so by substituting $\mathcal{Y} \mapsto \mathcal{P}(\mathcal{X})$ and $d_{\mathcal{Y}} \mapsto W_p$ in Equation 3, we can define the ($p$-)optimal transport dataset distance between datasets $\mathcal{D}_A$ and $\mathcal{D}_B$ by

$$\text{OTDD}(\mathcal{D}_A, \mathcal{D}_B) := \text{OT}_{(d_{\mathcal{X}}^p \times W_p^p)^{\frac{1}{p}}}(G(\mathcal{D}_A), G(\mathcal{D}_B)) \tag{4}$$

### A.1.2    COMPUTATIONAL CONSIDERATIONS

As we aim for a practical fine-tuning workflow, computational cost is a crucial concern. While Alvarez-Melis & Fusi (2020) proposed two variants of OTDD—the exact one and a Gaussian approximation, we observe from our experiments that optimizing the exact OTDD leads to better performance. In the following, we will focus on analyzing the computational cost of the exact OTDD.

Given datasets with $D$-dimensional feature vectors, estimating vanilla OT distances can be computationally expensive and has a worst-case complexity of $O(D^3 \log D)$ (Pele & Werman, 2009). However, adding an entropy regularization term $\epsilon H(\pi | \alpha \otimes \beta)$ to Equation 2, where $H$ is the relative entropy and $\epsilon$ controls the time-accuracy trade-off, can be solved efficiently with the Sinkhorn algorithm (Cuturi, 2013). This reduces OT's empirical complexity to $O(D^2)$ and makes the time cost for computing OTDD manageable for ORCA's workflow.

During implementation of ORCA, we also observed memory issues for computing OTDD using the entire target and source datasets on GPUs. To alleviate this, we propose a class-wise subsampling strategy for approximating OTDD on GPUs (Algorithm 1). In short, we split the $K$-class target dataset into $K$ datasets based on the labels and compute the class-wise OTDD between each single-class target dataset and the *entire source dataset*. Each class-wise OTDD can be approximated with the average of batch samples similar to how stochastic gradient descent approximates gradient descent. After that, we approximate the OTDD between the target and source datasets using the

---

**Algorithm 1** Efficient approximation of OTDD using class-wise subsampling.

---

**Input:** target dataset $\{x^t, y^t\}$, number of target classes $K^t$, source dataset $S = \{x^s, y^s\}$, sub-sample size $b$, subsample round $R$
**for** each class $i \in [K^t]$ in the target dataset **do**
    Compute class weight $w_i = \frac{\text{number of target data in class } i}{\text{total number of target data}}$
    Generate data loader $D_i$ consisting of data in class $i$
**end for**
**for** $i \in [K^t]$ **do**
    **for** $r \in [R]$ **do**
        Subsample $b$ target data points $D_{ir}$ uniformly at random from $D_i$
        Compute class-wise distance $d_{ir} = OTDD(D_{ir}, S)$
    **end for**
    Approximate class-wise OTDD by $d_i = \frac{1}{R} \sum_{i=1}^{R} d_{ir}$
**end for**
Approximate OTDD by $d = \sum_{i=1}^{K^t} w_i \cdot d_i$

---

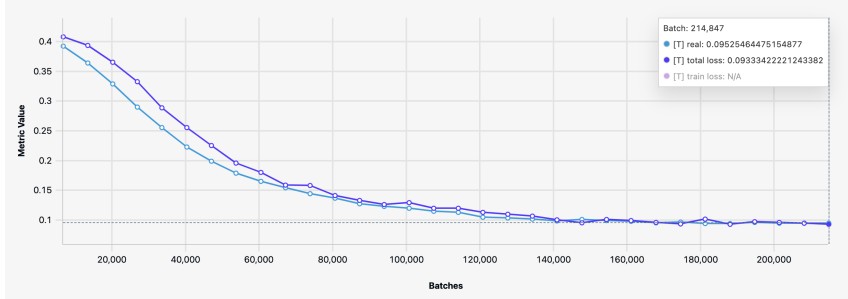

Figure 6: Screenshot of OTDD curves during embedding learning for the task ListOps. x-axis is the number of optimization steps, y-axis represents OTDD (1E-2). We use Algorithm 1 to approximate the exact OTDD as the loss function for optimization on GPU (purple curve). We also track the actual OTDD on CPU (blue curve). We can see that the proposed algorithm works well, which allows us to perform embedding learning efficiently.

weighted sum of the $K$ class-wise OTDDs. To verify that the approximation works empirically, we track the approximated OTDD (computed on GPUs) and the actual OTDD (computed on CPUs) and visualize the loss curves during ORCA's embedder learning process (Figure 6). We can see that the estimated value adheres to the actual value.

Leveraging both the Sinkhorn algorithm and class-wise approximation, the embedder learning process only takes up a small fraction of the total fine-tuning time in practice, as shown in Table 5. Hence, we invest a reasonable time budget but achieve significantly improved cross-domain transfer performance using ORCA.

Table 5: We record the runtime (in hours) of ORCA's embedding learning stage and the fine-tuning stage for each task. Then, we compute the ratio between the two. Averaged across tasks, embedding learning with OTDD only takes about 10% of the time needed for fine-tuning. All experiments are performed on NVIDIA V100 GPUs.

|  | CIFAR-100 | Spherical | Darcy Flow | PSICOV | Cosmic | NinaPro | FSD50K |
|---|---|---|---|---|---|---|---|
| Embedding | 1.6 | 1.8 | 0.18 | 0.35 | 0.2 | 0.3 | 0.21 |
| Fine-tuning | 9.2 | 9.3 | 0.86 | 1.1 | 3.5 | 1.1 | 12.5 |
| Embedding/Fine-tuning | 17% | 19% | 20% | 31% | 5% | 27% | 2% |

|  | ECG | Satellite | DeepSEA | JSB Chorales | ListOps | Homology |
|---|---|---|---|---|---|---|
| Embedding | 0.6 | 0.26 | 0.14 | 0.1 | 0.2 | 0.16 |
| Fine-tuning | 23.1 | 37.5 | 14.8 | 0.13 | 17.8 | 6.0 |
| Embedding/Fine-tuning | 3% | 1% | 1% | 76% | 1% | 2% |

## A.2 INFORMATION ABOUT EVALUATION TASKS

Table 6: Summary about each evaluation task and the hand-designed expert models used. The top 10 rows are for the 10 datasets in NAS-Bench-360 (Tu et al., 2022).

| Task name | # Data | Data dim. | Type | License | Learning objective | Expert arch. |
|---|---|---|---|---|---|---|
| CIFAR-100 | 60K | 2D | Point | CC BY 4.0 | Classify natural images into 100 classes | DenseNet-BC (Huang et al., 2017) |
| Spherical | 60K | 2D | Point | CC BY-SA | Classify spherically projected images into 100 classes | S2CN (Cohen et al., 2018) |
| NinaPro | 3956 | 2D | Point | CC BY-ND | Classify sEMG signals into 18 classes corresponding to hand gestures | Attention Model (Josephs et al., 2020) |
| FSD50K | 51K | 2D | Point (multi-label) | CC BY 4.0 | Classify sound events in log-mel spectrograms with 200 labels | VGG (Fonseca et al., 2021) |
| Darcy Flow | 1100 | 2D | Dense | MIT | Predict the final state of a fluid from its initial conditions | FNO Li et al. (2021a) |
| PSICOV | 3606 | 2D | Dense | GPL | Predict pairwise distances between residuals from 2D protein sequence features | DEEPCON (Adhikari, 2019) |
| Cosmic | 5250 | 2D | Dense | Open License | Predict propablistic maps to identify cosmic rays in telescope images | deepCR-mask (Zhang & Bloom, 2020) |
| ECG | 330K | 1D | Point | ODC-BY 1.0 | Detect atrial cardiac disease from a ECG recording (4 classes) | ResNet-1D (Hong et al., 2020) |
| Satellite | 1M | 1D | Point | GPL 3.0 | Classify satellite image pixels' time series into 24 land cover types | ROCKET (Dempster et al., 2020) |
| DeepSEA | 250K | 1D | Point (multi-label) | CC BY 4.0 | Predict chromatin states and binding states of RNA sequences (36 classes) | DeepSEA (Zhou & Troyanskaya, 2015) |
| JSB Chorales | 229 | 1D | Dense | CC BY-SA | Predict the next note from sheet music | Dilated TCN (Bai et al., 2018) |
| ListOps | 55K | 1D | Point | MIT | Model hierarchically structured data in a longcontext scenario | Reformer (Kitaev et al., 2020) |
| Homology | 12K | 1D | Point | | Predict the fold for a protein | LSTM (Rao et al., 2019) |

## A.3 EXPERIMENT DETAILS

Below, we summarize details for implementing ORCA and evaluating it on the selected 13 tasks. The code and configuration file for reproducing each experiment can be found in the supplementary material. We will also release ORCA's best checkpoint for each task later.

### A.3.1 PRETRAINED MODELS

We evaluated ORCA with two pretrained models in our experiments. In Table 2, for all 2D tasks including CIFAR-100, Spherical, Darcy Flow, PSICOV, Cosmic, NinaPro, and FSD50K, we use the following model. As Swin has a pretrained resolution, we reshape the inputs for our tasks to the resolution before feeding them into the model.

| Name | Pretrain | Resolution | Num Params | FLOPS | FPS |
|---|---|---|---|---|---|
| Swin-base (Liu et al., 2021c) | ImageNet-22K | 224×224 | 88M | 15.4G | 278 |

For all 1D tasks including ECG, Satellite, DeepSEA, JSB Chorales, ListOps,and Homology, we use the following model:

| Name | Pretrain | Num Params | FLOPS |
|---|---|---|---|
| RoBERTa-base (Liu et al., 2019b) | Five English-language corpora | 125M | 1.64E20 |

We use the Hugging Face transformers library Wolf et al. (2019) to implement the pretrained models.

### A.3.2 TASK DATA PREPARATION

For all the NAS-Bench-360 tasks, each dataset is preprocessed and split using the script available on `https://github.com/rtu715/NAS-Bench-360`, with the training set being used for hyperparameter tuning, embedding learning, and fine-tuning. We obtain the data processing script for JSB data from `https://github.com/locuslab/TCN`, for ListOps from `https://github.com/kzl/universal-computation`, and for Homology from `https://github.com/songlab-cal/tape`.

### A.3.3 HYPERPARAMETER TUNING

As ORCA is both task-agnostic and model-agnostic, it can be applied to fine-tuning a variety of pretrained transformers on drastically different end tasks with distinct datasets. Hence, it is hard to define one set of fine-tuning hyperparameters for all (model, task) pairs. At the same time, optimizing large-scale pretrained transformers can be challenging due to their large model sizes, as the downstream performance depends largely on the hyperparameters used. For instance, using a large learning rate can distort pretrained weights and lead to catastrophic forgetting. Therefore, in our experiments, given a (model, task) pair, we first apply hyperparameter tuning using the Asynchronous Successive Halving Algorithm (ASHA) (Li et al., 2020a) to the *standard fine-tuning setting* (i.e., after initializing the embedder and predictor architectures, directly updating all model weights to minimize the task loss) to identify a proper training configuration. Then, we use the same set of hyperparameters found for all our experiments for the particular (model, task) combination. Note that even though we did not explicitly state this in the main text, the hyperparameter tuning stage can be directly integrated into the ORCA workflow between stage 1 and stage 2. In this sense, ORCA is still an automated cross-modal transfer workflow that works for diverse tasks and different pretrained models.

The configuration space for ASHA is as follows:

- Batch size: 32, 128, 512, 1024 for Swin; 16, 56, 256, 512 for RoBERTa
- Optimizer: SGD, Adam, AdamW
- Learning rate: 1E-2, 1E-3, 1E-4, 1E-5, 1E-6
- Weight decay: 1E-2, 1E-3, 1E-4, 1E-5, 1E-6

Note that to fit each experiment on a single GPU, we set a fixed batch size (32 for Swin and 16 for Roberta) and vary the gradient accumulation step instead of actually varying the batch size, but the effect is the same.

### A.3.4 ORCA ONLY: EMBEDDING LEARNING WITH OTDD

After initializing the embedder architecture for each task, we train it to minimize the OTDD between the embedded target features and embedded source features.

For source datasets, we use CIFAR-10 for Swin and CONLL-2003 for RoBERTa. We sample 5000 data points to compute OTDD. In practice, we can pass the source data through the pretrained embedder once and save all the embedded features, so we don't have to pay the cost of obtaining the source features each time we fine-tune a new model.

For classification tasks, we directly use the labels provided by the end task to compute OTDD. For dense tasks, we perform K-Means clustering on the target data to obtain pseudolabels for OTDD computation. The number of clusters is set to the number of classes of the source dataset, e.g., 10 for 2D tasks that use CIFAR-10 as the source dataset.

To compute the embedding learning objective, we use the OTDD implementation of the original paper provided here: `https://github.com/microsoft/otdd`. As for the hyperparameters, we use the batch size, learning rate, optimizer, and weight decay obtained from A.3.3. The others are fixed across different tasks:

- Embedding learning epochs: 60
- Learning rate scheduler: decay by 0.2 every 20 epochs

### A.3.5 FINE-TUNING

Besides the searched hyperparameters, we also fix the following hyperparameters for fine-tuning.

- Fine-tuning epochs: 100 for Swin tasks, 60 for RoBERTa tasks
- Learning rate scheduler: we use the linear decay with min_lr = 0 and 5 warmup epochs

### A.3.6 TRAIN-FROM-SCRATCH

This baseline is trained using the same hyperparameter configuration (number of epochs, batch size, learning rate, etc) as the fine-tuning baseline.

### A.3.7 EVALUATION

When training/fine-tuning is finished, we evaluate the performance of all models following the NAS-Bench-360 protocol. We first report results of the target metric for each task by running the model of the *last* epoch on the test data. Then, we report aggregate results via performance profiles (Dolan & Moré, 2002), a technique that considers both outliers and small performance differences to compare methods across multiple tasks robustly. In such plots, each curve represents one method. The $\tau$ on the $x$-axis denotes the fraction of tasks on which a method is no worse than a $\tau$-factor from the best. The performance profile for our experiments is shown in Figure 2.

### A.4 ABLATION STUDY ON EMBEDDING LEARNING METRICS

As motivated in Section 4.2.1, we present here an ablation study on the embedding learning metrics that we have considered for minimizing distribution dissimilarity. The results show that (1) performing feature alignment generally helps downstream adaptation, regardless of which metric we minimize; (2) OTDD leads to the best overall performance, so we chose it for our workflow. Our findings confirm that it is the general idea of data alignment, rather than a specific metric, that makes cross-modal transfer work.

Specifically, we experiment with OTDD, maximum mean discrepancy (MMD) (Gretton et al., 2012), the mutual-information-based TransRate (Huang et al., 2022), and (pairwise) Euclidean distance. We learn the embedders to minimize these metrics and then fine-tune the pretrained models. The test errors are as follows.

Table 7: Prediction errors of different distance metrics. OTDD achieves the best overall performance. "w/o Emb. Learning" represents fine-tuning without embedder learning.

| | CIFAR-100 | Spherical | Darcy Flow | PSICOV | Cosmic | NinaPro | FSD50K |
|---|---|---|---|---|---|---|---|
| OTDD | **6.53±0.079** | **29.85±0.72** | **7.3E-3±6.8E-5** | 1.91±0.038 | **0.21±0.0050** | 7.54±0.39 | **0.56±0.013** |
| MMD | 6.62±0.092 | 33.64±2.57 | 7.4E-3±3.4E-4 | **1.9±0.016** | 0.32±0.054 | **7.48±0.23** | 0.58±0.004 |
| TransRate | 6.66±0.084 | 30.32±0.53 | 8.0E-3±6.5E-5 | 1.92±0.03 | 0.31±0.0037 | 8.64±0.26 | 0.57±0.01 |
| Euclidean | 7.09±0.48 | 32.33±2.03 | 7.3E-3±1.9E-4 | 1.91±0.019 | 0.27±0.021 | 7.51±0.11 | 0.59±0.02 |
| w/o Emb. Learning | 7.67±0.55 | 55.26±1.63 | **7.3E-3±1.1E-4** | 1.92±0.039 | 0.24±0.0080 | 8.35±0.75 | 0.63±0.014 |

| | ECG | Satellite | DeepSEA | JSB Chorales | ListOps | Homology | |
|---|---|---|---|---|---|---|---|
| OTDD | **0.29±0.0052** | 11.59±0.18 | **0.29±0.006** | **2.44±0.056** | **51.90±2.18** | **87.21±0.50** | |
| MMD | 0.40±0.018 | **11.29±0.087** | 0.38±0.077 | 2.57±0.045 | 63.94±1.3 | 89.3±0.57 | |
| TransRate | 0.44±0.021 | 11.35±0.24 | 0.38±0.077 | 2.56±0.046 | 63.02±3.37 | 97.21±0.00 | |
| Euclidean | 0.41±0.009 | 11.4±0.078 | 0.34±0.002 | 2.56±0.043 | 60.2±0.41 | 89.97±1.18 | |
| w/o Emb. Learning | 0.44±0.0056 | 13.86±1.47 | 0.51±0.0001 | 2.59±0.098 | 72.79±1.96 | 89.31±0.60 | |

## A.5 RUNTIME OF ORCA VS. FPT

In Table 3, we compare with the FPT setting, which only fine-tunes the layer norms of the pretrained transformer models. As we have shown already, the downstream performance of fine-tuning only a subset of the parameters is less competitive than fine-tuning all parameters. Below, we show that the time saved for updating only layer norms is also not that significant. Therefore, we suggest performing full fine-tuning when time and computational resources allow.

Table 8: We record the total runtime (in hours) for four settings: ORCA with full fine-tuning, ORCA with tuning layer norms, full fine-tuning (without embedding learning), and fine-tuning layer norms (FPT). We can see that tuning the layer norms does not bring significant benefit in terms of reducing the model development time, but it sacrifices the downstream performance of the resulting models.

| | CIFAR-100 | Spherical | Darcy Flow | PSICOV | Cosmic | NinaPro | FSD50K |
|---|---|---|---|---|---|---|---|
| ORCA | 10.8 | 11.1 | 1.04 | 1.45 | 3.7 | 1.4 | 12.71 |
| ORCA (layernorm) | 8.7 | 8.9 | 0.76 | 1.15 | 3.5 | 1.0 | 8.96 |
| Fine-tuning | 9.2 | 9.3 | 0.86 | 1.1 | 3.2 | 1.1 | 12.5 |
| Fine-tuning (layernorm) | 7.1 | 7.1 | 0.58 | 0.8 | 3.0 | 0.7 | 8.75 |

| | ECG | Satellite | DeepSEA | JSB Chorales | ListOps | Homology | |
|---|---|---|---|---|---|---|---|
| ORCA | 23.7 | 37.76 | 14.94 | 0.23 | 18.0 | 6.16 | |
| ORCA (layernorm) | 18.0 | 25.56 | 11.24 | 0.2 | 13.2 | 4.56 | |
| Fine-tuning | 23.1 | 37.5 | 14.8 | 0.13 | 17.8 | 6.0 | |
| Fine-tuning (layernorm) | 17.4 | 25.3 | 11.1 | 0.1 | 13.0 | 4.4 | |

## A.6 PROMPTING

Apart from fine-tuning, a new paradigm of working with large-scale pretrained models is prompting, i.e., we do not update the pretrained weights but only modify the input and query the model for the desired output. Existing language prompting methods (e.g., Liu et al., 2022) are generally not suitable for cross-modal learning due to the difficulty of designing natural prompts for diverse data types. For the 1D tasks we study, there is even no notion of "discrete tokens." Another line of work studies visual prompting by modifying 2D inputs for querying vision transformers. We test two such algorithms, VP (Bahng et al., 2022) and VPT (Jia et al., 2022), on three classification tasks in our task suite. They are not applicable to the remaining tasks because either the inputs cannot be reshaped to look like images or the outputs are not classification logits.

We test VPT with the pretrained Swin-Base Transformer (the same model we used for ORCA) and VP with the pretrained ResNet-50 (as the official implementation does not support vision transformers). The results are shown in Table 9. In general, prompt tuning is less effective than fine-tuning, and the two baselines perform significantly worse than ORCA. This is not surprising given that prompting methods are more intuitively suited to in-modality transfer, where the target and the source data have similar structure or semantic meaning. However, when the target data (e.g., electromyography signals, as in the NinaPro dataset) is drastically different from image data, it is difficult to design prompts or expect good performance by only modifying the inputs without fine-tuning the pretrained models.

Table 9: Prediction errors of ORCA vs. visual prompting methods.

| | Spherical | NinaPro | ECG |
|---|---|---|---|
| ORCA | **29.85±0.72** | **7.54±0.39** | **0.29±0.0052** |
| VP | 98.05±0.13 | 33.18±0.23 | 0.57±0.0044 |
| VPT | 49.53±1.45 | 31.46±0.83 | 0.40±0.016 |

## A.7 COMPATIBILITY WITH IN-MODALITY TRANSFER

A natural question to ask is whether ORCA can also tackle in-modality tasks. While we design ORCA to enable cross-modal transfer, we hypothesize that it should facilitate same-modality transfer if two domains have large dataset distance. To validate this, we test ORCA on DomainNet

Table 10: We use the dataset splits in Tan et al. (2020), which removed some mislabeled outliers, and report the prediction errors for ORCA and fine-tuning (using Swin-base).

|  | Real | Painting | Sketch | Clipart |
|---|---|---|---|---|
| ORCA | **96.71±0.02** | **94.71±0.13** | **94.93±0.24** | **93.61±0.54** |
| Fine-tuning | 93.33±1.33 | 75.79±0.86 | 83.00±0.13 | 86.01±2.62 |

datasets, which are commonly used to evaluate homogeneous DA methods (Peng et al., 2019). From Table 10, we can see that ORCA achieves significantly better performance than the fine-tuning baseline, which shows that the feature matching of ORCA can also help in-domain generalization.

