# OpenReview forum: "Tackling Diverse Tasks via Cross-Modal Transfer Learning"
_ICLR.cc/2023/Conference — Submitted to ICLR 2023_

### Official Review · Reviewer_tx7E · 2022-10-21

**Confidence:** 4
**Correctness:** 2
**Technical Novelty And Significance:** 2
**Empirical Novelty And Significance:** 2
**Recommendation:** 5

**Clarity, Quality, Novelty And Reproducibility:**

The paper is not clear and hard to follow. This work is incremental and lacks novelty.

**Strength And Weaknesses:**

Strengths:
- The authors have evaluated the proposed method on many different datasets.
- The motivation of this work is interesting.

Weaknesses:
- The presentation of the paper is not clear. For example, Figure 1 cannot give us clear information (e.g., how does ORCA work? What represents cross-modal transfer?)
- This work lacks novelty. It utilizes the existing techniques to fine-tune models on different datasets, which is incremental.
- Although this paper evaluates the proposed method on many different datasets, it lacks sufficient comparison experiments with state-of-the-art baselines.
- The authors claim their method could achieve task-specific adaption. Why does it need fine-tune? If it could achieve task-specific adaption on the upstream pre-training, it could achieve good performance on the downstream tasks without finetuning.

**Summary Of The Paper:**

In this paper, the authors study the cross-modal transfer ability of large-scale pre-trained models. The authors present a general workflow to fast and automatically exploit existing pre-trained models for different tasks. The authors provide some experiments to evaluate the proposed method.

**Summary Of The Review:**

The paper is not clear and hard to follow. This work is incremental and lacks novelty. The experiments are not solid for comprehensively evaluating the proposed method.

---

> ### Author Response · Authors · 2022-11-12
> **Response to Reviewer tx7E**
>
> Thank you for your feedback. We respectfully disagree with the claim that “this work is incremental and lacks novelty.” Our novelty is grounded in the problem we study, the viable workflow we propose, and its empirical success. Please see the first part of the general response for a detailed explanation. Also, we believe that obtaining drastically better results relative to recently published works at top conferences using an intuitive and simple approach is a significant but not incremental technical contribution. This is echoed by reviewer AGVC: *“this paper tackles this challenging and important problem and gives a simple but effective solution.”*
>
> Please find our response to your other questions below.
>
> &nbsp;
>
> **_1. “The presentation of the paper is not clear. For example, Figure 1 cannot give us clear information (e.g., how does ORCA work? What represents cross-modal transfer?)”_**
>
> Could you provide us with concrete suggestions on how to improve the presentation of the paper, as all the other reviewers agree that we have presented the key idea with clarity? We would be happy to incorporate them in revision.
>
> Currently we think Figure 1 has provided a thorough description of how ORCA works, especially since we have supplemented the illustration with text explanations in the caption. **Stage 2: Learn Task Embedder** represents how we enable cross-modal transfer via mapping the distribution of the target task data to that of the pretraining data.
>
> &nbsp;
>
> **_2. “This work lacks novelty. It utilizes the existing techniques to fine-tune models on different datasets, which is incremental.”_**
>
> We believe this is an unfair and reductive characterization of the work. We are the first to introduce the optimal transport dataset distance as an embedding learning objective to the transfer learning setting. Though the metric itself is not new, how we utilize it to enable significant empirical success in cross-modal transfer is new. A new combination or application of existing techniques is inherently novel [1]. Also note that from the new ablation study we provide, we can see that **OTDD is more of a plug-and-play design choice** and that **ORCA’s effectiveness comes mostly from the data alignment process we propose**, but not the exact metric used for data alignment.
>
> &nbsp;
>
> **_3. “it lacks sufficient comparison experiments with state-of-the-art baselines.”_**
>
> We are not quite sure what specific methods you are referring to. Would you be willing to provide us with more information as we would be happy to investigate more? Also, **please note that we have added additional baselines in the general response to address your question**, such as training transformers from scratch and prompting-based methods. We have shown that ORCA outperforms them as well.
>
> &nbsp;
>
> **_4. “The authors claim their method could achieve task-specific adaption. Why does it need fine-tune?”_**
>
> Fine-tuning is necessary for two reasons. First, as shown in Stage 1 of Figure 1, apart from the embedder, we also need a **predictor** to map the output of the transformer (sequence vectors) to the task output (e.g., classification logits, dense maps). This predictor is required even for traditional in-modality transfer because we might have different labels for different downstream tasks. As the prediction layers are initialized randomly and must be learned from scratch (similar to the linear probing setting), we need fine-tuning to learn the predictor for inference. Second, as discussed in Section 3.3, the fine-tuning stage can help us further align the embedder and predictor with the pretrained model to improve downstream performance.
>
> To support the above points, we run an additional set of experiments that fine-tune only the prediction head after learning the embedder. Results show that this can already achieve better accuracy than many AutoML methods and Perceiver IO in some tasks. (Note that in this case, we do not update the weights of the attention blocks at all, even though they are pretrained in an entirely different domain.) This shows that ORCA’s data alignment step results in a significant amount of task-specific adaptation. However, the performance of fine-tuning only the prediction head still lags behind ORCA. This supports our second claim that full fine-tuning can improve downstream performance by further aligning the embedder with the pretrained model.
>
> | |CIFAR-100 |Spherical |Darcy Flow|PSICOV |Cosmic|NinaPro|FSD50K|ECG |Satellite|DeepSEA|JSB Chorales| ListOps| Homology|
> |:-:|:-:|:-:|:-:|:-:|:-:|:-:|:-:|:-:|:-:|:-:|:-:|:-:|:-:|
> |OTDD|**6.53**|**29.85**|**7.3E-3**|**1.91**|**0.21**|**7.54**|**0.56**|**0.29**|**11.59**|**0.29**|**2.44**|**51.9**|**87.21**|
> |OTDD (predictor only)|9.07|58.99|2.4E-1|12.54|0.49|20.03|0.68|0.44|16.31|0.34|29.22|67.77|90.48|
>
> &nbsp;
>
> **References**
>
> [1] Michael J. Black, Novelty in Science: A Guide for Reviewers, Perceiving Systems Blog, https://perceiving-systems.blog/en/news/novelty-in-science

---

> > ### Comment · Area_Chair_Q3hx · 2022-11-25
> > **Please update your review by today**
> >
> > Since your rating is different from others and authors provide detailed responses. May I know whether these responses change your mind? If yes, please update your rating score accordingly. If no, please state your concerns ASAP?
> >
> > Best,
> > AC

---

> ### Author Response · Authors · 2022-11-22
> **Looking Forward to Further Feedback**
>
> We appreciate the reviewer's time and effort in providing valuable suggestions for our work. We have updated the submission to include more baselines and clarify our presentation as suggested. Please let us know if our response has addressed your concerns. We would be happy to answer any other questions you may have.

---

### Official Review · Reviewer_J4V2 · 2022-10-22

**Confidence:** 3
**Correctness:** 3
**Technical Novelty And Significance:** 3
**Empirical Novelty And Significance:** 3
**Recommendation:** 5

**Clarity, Quality, Novelty And Reproducibility:**

Quality: The paper is likely to have a modest impact on the community. Clarity: The paper is well organized but the presentation has minor details that could be improved. Originality: The main ideas of the paper are not novel or have limited novelty. Please see the weaknesses.

**Strength And Weaknesses:**

Strengths:
In the big model era, how to effectively adapt pre-trained models to different downstream tasks is an interesting and practical problem. Therefore, studying the cross-modal transfer ability of large-scale pre-trained models is significant.

Weaknesses:
1. The key idea of the proposed method is resorting to the OTDD between the proxy source data and target data to fine-tune the given pre-trained model. However, the definition and usage of OTDD are not clear enough. More importantly, according to the unclear statement, OTDD seems to be a modified version of OT which introduces the label information for complimentary. From this perspective, the reviewer wonders how the OTDD works under the scenario where the source and target data have no share labels.
2. The authors argue that the proposed ORCA is an effective transfer learning method for pre-trained models compared to vanilla fine-tuning and hand-craft prompt tuning. Although the empirical results on two popular vision and language models (Swin and Roberta) show the superiority of the proposed method compared to the vanilla baselines. The effectiveness of ORCA should be further defended by comparing it to the recently-proposed vision and language prompt tuning methods such as [1].
[1] Visual Prompting: Modifying Pixel Space to Adapt Pre-trained Models

**Summary Of The Paper:**

This paper studies an pratical problem that how to effectively transfer pretrained models for diverse tasks. To this end, the authors proposed ORCA, which conducts task-specific adaptation by minimizing the OTDD between the proxy source embeddings and target embeddings. Extenvise transferring experiments on different tasks have been conducted to verify the effectiveness of the proposed ORCA.

**Summary Of The Review:**

Please see weaknesses

---

> ### Author Response · Authors · 2022-11-12
> **Response to Reviewer J4V2**
>
> Thank you for your feedback. We respectfully disagree with the claim that “the main ideas of the paper are not novel.” Our novelty is grounded in the problem we study, the viable workflow we propose, and its empirical success. Please see the first part of the general response for a detailed explanation. Also note that in the new ablation study we provide in the general response, we showed that **OTDD is more of a plug-and-play design choice** and that **ORCA’s effectiveness comes mostly from the data alignment process we propose**, but not the exact metric used for data alignment. In general, we believe that obtaining drastically better results relative to recently published works at top conferences using an intuitive and simple approach is a significant contribution. This is echoed by reviewer AGVC: *“this paper tackles this challenging and important problem and gives a simple but effective solution.”*
>
> Please find our response to your other questions below.
>
> &nbsp;
>
> **_1. “However, the definition and usage of OTDD are not clear enough.”_**
>
> Thank you for pointing this out. Due to space limitations, we only explained how OTDD accounts for the label information in Section 3.2, which is its key difference from the classic OT distance. We deferred a more detailed exposition on OTDD and its usage to Section A.1 of the Appendix. However, we are aware that readers unfamiliar with OTDD can have trouble understanding how it helps embedding learning, so we will revise the paper to make the definition and usage clearer.
>
> &nbsp;
>
> **_2. “the reviewer wonders how the OTDD works under the scenario where the source and target data have no share labels.”_**
>
> As noted in Section 3.2, OTDD “works even when the label sets are disjoint or completely unrelated” (e.g., digits to letters). This is because OTDD views the label sets from a purely geometric perspective by modeling each label as a distribution over the feature vectors in that class. In this sense, **the semantic meaning of the label does not actually matter. It is the feature distribution of samples that belong to the label’s class that characterizes the label.** We will update the paper to make this point clearer. We also refer the reviewer to Section 4 of the OTDD paper for a detailed explanation.
>
> &nbsp;
>
> **_3. “The effectiveness of ORCA should be further defended by comparing it to the recently-proposed vision and language prompt tuning methods such as [1]”_**
>
> We appreciate your suggestion, and **we have included two visual prompt tuning methods, VP [1] and VPT [2], as baselines in the general response**. We tested these methods on three 2D tasks and observed that for cross-modal tasks, prompt tuning is in general less effective than fine-tuning, and **the two baselines perform significantly worse than ORCA**. This is not surprising given that prompting methods are more intuitively suited to in-modality transfer, where the target and the source data have similar structure or semantic meaning. However, when the target data (e.g., electromyography signals, as in the NinaPro dataset) is drastically different from image data, it is difficult to design prompts or expect good performance by only modifying the inputs and not fine-tuning the pretrained models. We did not test language prompting methods for 1D tasks due to the same reason—for most of the tasks we evaluate, there is no definition of "discrete tokens," so prompting makes less sense and many methods are just not applicable.
>
> &nbsp;
>
> **References**
>
> [1] Bahng et al., Visual Prompting: Modifying Pixel Space to Adapt Pre-trained Models, 2022.
>
> [2] Jia et al., Visual Prompt Tuning, ECCV 2022.

---

> > ### Comment · Reviewer_J4V2 · 2022-11-25
> > **Reply**
> >
> > Although the authors have provided additive experiments and explainations, the reviewer still concerns about the technical novelty of this paper. After reading the revised paper again, considering the novelty and clearness, the reviewer decides to keep the score of first round.

---

> > > ### Author Response · Authors · 2022-11-25
> > > **Asking for clarification**
> > >
> > > Hi Reviewer J4V2. In your first (since-deleted) reply, you seemed satisfied with our response to the two weaknesses you mentioned in the initial review; we clarified the use of OTDD and compared with additional baselines, including the prompting methods you requested, to which you replied "*the issue has been well addressed, and I am willing to raise my rating*" on Nov 21. However, in this response, you express concern about "technical novelty," but (1) without providing specific examples or context, so we are unsure what you are referring to, and (2) this has already been well-addressed in our general response, where we emphasize that though the metric itself is not new, how we utilize it to enable significant empirical success in cross-modal transfer is new. Most of the reviewers seem to acknowledge this point after our rebuttal.
> > >
> > > You have also edited your original review to say "*The main ideas of the paper are not novel or have limited novelty. Please see the weaknesses.*" But as of this writing, you have listed no weaknesses related to novelty, only to clarity and baselines, both of which we have addressed to your satisfaction. Could you please provide evidence for what specific concerns you have so that we can address them?

---

> > > ### Author Response · Authors · 2022-12-05
> > > **Looking forward to further feedback**
> > >
> > > We appreciate the reviewer's time and effort in providing valuable suggestions for our work. As mentioned in our last comment (titled "asking for clarification"), we would be grateful if you could inform us of what *specific* concerns you have so that we can provide a better answer. If we appear to have addressed all of your other concerns, we would like to ask the reviewer to consider raising their score.

---

### Official Review · Reviewer_WPBn · 2022-10-24

**Confidence:** 4
**Correctness:** 4
**Technical Novelty And Significance:** 2
**Empirical Novelty And Significance:** 2
**Recommendation:** 6

**Clarity, Quality, Novelty And Reproducibility:**

The key idea and the pipeline are described clearly. According to the detailed instructions, I believe that this idea is easy to reproduce. However, it lacks enough novelty since the whole pipeline is build on previous techniques.

**Strength And Weaknesses:**

Strength
1. ORCA achieves good performance compared to other approaches on 13 tasks with different modalities, input dimensions, and prediction types.
2. The importance of matching feature distribution is well supported and it is compatible with the in-modality transfer.

Weaknesses
1. The technical contribution is not significant. Though the performance is good and the effect of OTDD is verified, the main idea of using OTDD as a target for data distribution mapping is somewhat trivial.
2. The custom embedding network uses a convolution layer for patch embedding like Vit. Does it only process image-like data? If so, it is inconsistent with the state as claimed before.
3. What is the effect of other metrics in Stage I?

**Summary Of The Paper:**

This paper focuses on cross-modal transfer learning that aims to utilize the pre-trained models for downstream tasks with diverse modalities. It devises a task specific embedding network to map data distribution to the pertained modality using the optimal transport dataset distance metric. Extensive experiments verify the efficacy of the proposed ORCA.

**Summary Of The Review:**

Though the proposed method achieves good performance on a range of tasks, it is an integration of existing techniques, which weakens the contribution of this paper.

---

> ### Author Response · Authors · 2022-11-12
> **Response to Reviewer WPBn**
>
> Thank you for your feedback. We respectfully but strongly disagree with the argument that ORCA is not significant because “the idea is somewhat trivial." As you stated in the strengths section, you agree that our method is indeed SOTA on an interesting problem, and we believe that **all else being equal, simpler is better**. Please find our detailed response below.
>
> &nbsp;
>
> **_1. “The technical contribution is not significant…the main idea of using OTDD as a target for data distribution mapping is somewhat trivial.”_**
>
> We believe this is an unfair and reductive characterization of the work. Our novelty is grounded in the problem we study, the viable workflow we propose, and its empirical success. Please see the first part of the general response for a detailed explanation. Also note that from the new ablation study we provide, we can see that **OTDD is more of a plug-and-play design choice** and that ORCA’s effectiveness comes mostly from the data alignment process but not the exact metric used for data alignment.
>
> In general, obtaining drastically better results relative to recently published works at top conferences using an intuitive and simple approach is a significant, non-trivial technical contribution. This is echoed by reviewer AGVC: *“this paper tackles this challenging and important problem and gives a simple but effective solution.”*
>
> &nbsp;
>
> **_2. “The custom embedding network uses a convolution layer for patch embedding like Vit. Does it only process image-like data? If so, it is inconsistent with the state as claimed before.”_**
>
> No, it does not only process image data. ORCA uses *different kinds* of convolutional layers in the embedder to process *different kinds* of data, e.g., conv1d for 1D data and conv2d for 2D data. In fact, most of the data that we consider are not image-like, but using convolution allows us to transform them into sequence vectors while preserving the intrinsic features of different data types effectively, as shown in previous works.
>
> For instance, the Temporal Convolutional Network [1] uses 1D convolutions to solve sequential MNIST, music modeling, and word-level language modeling problems, outperforming LSTMs and RNNs. The Structured State Spaces (S4) model [2] and SGConv [3] use 1D convolutions to achieve SOTA results on speech classification and the Long Range Arena benchmark [4]. Moreover, recent NAS work [5] has shown that equipping ResNets with various types of convolutions can result in highly effective models for all NAS-Bench-360 tasks (we compare ORCA with this method, DASH, in Table 2). Therefore, we use convolutions in the embedders for all tasks because: (1) they have been shown to be effective for feature extraction; (2) we want a unified way of defining the embedder architecture for diverse tasks, and using convolutions of different dimensions allows us to do so.
>
> Thank you for asking this question. We will clarify this in the revised paper. We would be happy to discuss more if you have further concerns.
>
> &nbsp;
>
> **_3. “What is the effect of other metrics in Stage I?”_**
>
> Are you referring to the metric for measuring distribution dissimilarity (OTDD) in **Stage 2**: Learning Task Embedder? If so, **we have added an ablation study on the effect of different metrics in the general response.** We experimented with OTDD, MMD, TransRate, and Euclidean distance. We chose OTDD for ORCA because it has better empirical performance. We would be happy to discuss more if you have other concerns.
>
> &nbsp;
>
> **References**
>
> [1] Bai et al., An Empirical Evaluation of Generic Convolutional and Recurrent Networks for Sequence Modeling, 2018.
>
> [2] Gu et al., Efficiently Modeling Long Sequences with Structured State Spaces, ICLR 2022.
>
> [3] Li et al., What Makes Convolutional Models Great on Long Sequence Modeling? 2022.
>
> [4] Tay et al., Long Range Arena: A Benchmark for Efficient Transformers, 2020.
>
> [5] Shen et al., Efficient Architecture Search for Diverse Tasks, NeurIPS 2022.

---

> > ### Comment · Reviewer_WPBn · 2022-11-21
> > **Response to the contribution**
> >
> > Thanks for the response. I have reviewed all the comments and realized I should adjust my judgment for this work. Although the proposed workflow does not bring too many new things from the perspective of detailed technical contributions (e.g., dimension transform, data alignment, and fine-tuning are generally used in other fields), the authors give sufficient experimental results to demonstrate the effectiveness of this conceptually simple framework. Therefore, I raised my rating.

---

### Official Review · Reviewer_AGVC · 2022-10-25

**Confidence:** 3
**Correctness:** 3
**Technical Novelty And Significance:** 3
**Empirical Novelty And Significance:** 3
**Recommendation:** 8

**Clarity, Quality, Novelty And Reproducibility:**

Clarity, quality: The paper is well organized and well written.
Novelty: As pre-trained models become increasingly powerful, there exists a desire to apply these models to aid the learning of downstream tasks. But the mismatch of data distribution from different modalities hinders the application. This paper tackles this challenging and important problem and gives a simple but effective solution.
Reproducibility: The code and configuration file for reproducing each experiment can be found in the supplementary material.

**Strength And Weaknesses:**

Strength:
As pre-trained models become increasingly powerful, there exists a desire to apply these models to aid the learning of downstream tasks. But the mismatch of data distribution from different modalities hinders the application. This paper tackles this challenging and important problem and gives a simple but effective solution. The solution of this paper leverages the property of transformer that it has a standard input structure without any inductive bias of data, and proposes to learn an embedder that can map any input to match such structure. To better align different modalities, the authors further use optimal transport to match the source and target datasets. Although being simple, this adaptation scheme is reasonable and the authors give enough empirical evidence of that this approach is effective.
The experiments show that ORCA performs generally better than SOTA methods.

Weakness:
One concern is the use of pre-trained model, which serve as a potential unfair comparison with other methods. In particular, the pretrained model the author chooses is RoBERTa and SwinTransformer trained on Five English-language corpora and ImageNet-22K, respectively. This may differ from other methods which potentially use different datasets for pre-training. So can authors provide some comparisons with state-of-the-art methods trained on the same dataset using identical architecture? I understand that such large-scale experiments are difficult to reimplement, but such comparisons would be greatly appreciated and would bolster the work.

**Summary Of The Paper:**

This paper introduces a new approach, ORCA, to tackle the cross-model transfer problem. ORCA equips a pre-trained transformer with a task-specific embedder and a task-specific predictor. When facing a downstream task, the embedder is learned to map data of any dimension to a standard sequence of tokens that is then fed into the pre-trained transformer. The predictor is learned to map the output to the specific label space. The learning objective is an optimal transport dataset distance that is used to match the source and target data distribution. After matching, the whole network is finetuned using standard loss.

**Summary Of The Review:**

Overall, this paper tackles the important and challenging cross-modal transfer problem and gives a feasible solution, despite lack of novelty and potentially unfair comparisons with other methods. Given that the problem and the proposed approach would be interesting to ICLR audience, I would tend to accept this paper.

---

> ### Author Response · Authors · 2022-11-12
> **Response to Reviewer AGVC**
>
> Thank you for your positive review. We hope to address your questions below.
>
> &nbsp;
>
> **_1. “can authors provide some comparisons with state-of-the-art methods trained on the same dataset using identical architecture?”_**
>
> We are not quite sure what exact methods you are referring to, so we respond from two perspectives that you might be interested in to show that our current comparison is fair.
>
> **1.1 Did we compare with other transfer learning methods that use different pretraining datasets?**
>
> If your concern is that ORCA’s success is due to the language corpora or ImageNet used to pretrain the models rather than our data alignment process, then please note that the standard fine-tuning baseline in Table 3 uses the exact same transformers pretrained with the same data (which meets your criteria of “state-of-the-art methods trained on the same dataset using identical architecture”). However, this baseline is consistently worse than ORCA, showing that aligning feature distributions through embedding learning, which is the key novelty of our work, is necessary for successful cross-modal transfer. In addition, we are not aware of any other cross-modal transfer methods that apply to the considered tasks but use different source datasets. We would be happy to investigate more if you could direct us to relevant literature.
>
> **1.2 Did we compare with models that have similar size and architecture?**
>
> If your concern is that RoBERTa and Swin have more parameters than hand-crafted and AutoML models, please note that **we have added a new train-from-scratch baseline in the general response** that trains RoBERTa and Swin using only the target data. In this case, the same architecture is used, but ORCA still significantly outperforms this baseline, demonstrating that there are indeed benefits to transferring knowledge from a well-studied domain instead of learning from scratch.
>
> We would be happy to discuss more if you want to clarify this question or have other concerns.

---

> > ### Comment · Reviewer_AGVC · 2022-11-21
> > **Reply to unfair comparison**
> >
> > Thanks for your answer to my concern of the unfair comparison. This issue has been well addressed, and I am willing to raise my rating.

---

### Official Review · Reviewer_uNsA · 2022-10-31

**Confidence:** 4
**Correctness:** 3
**Technical Novelty And Significance:** 4
**Empirical Novelty And Significance:** 3
**Recommendation:** 8

**Clarity, Quality, Novelty And Reproducibility:**

The paper is well-written 3/4,
clarity/easy to understand 2/4, some ideas needs better reasoning
and novelty 3/4.

**Strength And Weaknesses:**

Strengthes
The paper proposes an general/universal approach to do cross modal transfer learning, by adding a dimension transformation embedding layer and pretrain it using OTDD
The paper describe the motivation, challenges and solution clearly, and there are enough empirical results supporting the claim.
The comparisons against other cross modal transfer solution is convincing, e.g from NAS based class, hand-crafted expertise class.
The paper did some deep dives into the effectiveness of transformation embedding pretaining.

Weaknesses
Some of the empirical results and statements requires better reasoning, e.g:
Why just pretraining the embedding transformation layers will make the final result even better than hand-crafted expertise modes by a big margin in some downstream tasks? like Spherical and JSB Chorales.
Why using OTDD when pretraining the transformation embedding layers, the metric and semantic goal is not connected explicitly.

There are some places requires more deep diving, e.g:
Using convolutional network to do dimension transferring from target X -> source X, using it for Orca-swin makes sense because they both belong to vision modeling, but why still use it for Orca-roberta? Why convolutional network is selected, and how does it compare to simple feed-forward network for dimension transformation, their comparison in down stream tasks, etc, to demonstrate the effectiveness of this choice

It is good to show both Orca-swin/roberta results in Table 2, let people understand the impact of pretrained body as well.

**Summary Of The Paper:**

This paper shows an universal approach of handling transfer learning in terms of cross-modal tasks, by using pre-training the embedding layers using OTDD loss metric. This paper although conduct empirical study on 13 tasks, demonstrating the effectiveness of embedding layer pretraining in cross modality learning. This approach is universal, effective and beats counterparts like NAS solutions, handcraft models, and other cross modal transfer tasks by quite some margin.

**Summary Of The Review:**

This is a good empirical paper that the idea makes quite sense, and the authors conduct thorough experiments to support the claim such that pretraining embedding layer served as cross modal transfer is very effective.

---

> ### Author Response · Authors · 2022-11-12
> **Response to Reviewer uNsA**
>
> Thank you for your positive feedback. We hope to respond to your questions below.
>
> &nbsp;
>
> **_1. “Some of the empirical results and statements requires better reasoning, e.g: Why just pretraining the embedding transformation layers will make the final result even better than hand-crafted expertise modes”_**
>
> Pretraining the embedding layers allows us to align the target feature distribution with that of the pretraining data. Only after this step can we effectively leverage the pretrained weights. And we do see that this alignment step helps utilizing cross-domain knowledge, as training RoBERTa from scratch outperforms fine-tuning *without* embedding learning but underperforms fine-tuning *with* embedding learning (ORCA) on many tasks (**see the new train-from-scratch results in the general response**). Furthermore, optimizing directly for the task loss without pretraining the embedding layers may distort the pretrained weights and lead to suboptimal solutions [1, 2], which may explain why ORCA performs so well.
>
> As for why ORCA can even outperform hand-crafted expert models, we see two reasons.
> - Hand-crafted models are developed by domain experts using domain knowledge. However, since it is hard for domain experts to always keep up with the latest progress in deep learning architectures, many hand-crafted models still use traditional architectures like CNNs and LSTMs, which can be less effective than larger and carefully-tuned transformers. ORCA was developed exactly to relieve the burden of domain experts and allow them to obtain high-performing models without having to design new architectures themselves.
> - In deep learning, the trend also seems to be that more data trumps domain knowledge [3]. Our approach effectively simulates a large-data setting by allowing us to exploit data from other modalities.
>
> We will add the above discussion to the experiment section of the paper.
>
> &nbsp;
>
> **_2. “Why using OTDD when pretraining the transformation embedding layers, the metric and semantic goal is not connected explicitly”_**
>
> We use OTDD because it has better empirical performance than other metrics such as MMD, TransRate, and Euclidean distance (**please check out our general response for the ablation study on these metrics**).  As for the semantic reason, OTDD “yields a transportation cost between datasets,” according to the OTDD paper. Consequently, learning the embedding layers to minimize OTDD is the same as learning how we can manipulate the target data distribution to look like the pretraining data distribution. Then, during fine-tuning, we can adapt the pretrained weights more easily without the risk of catastrophic forgetting. We will clarify this semantic connection in the revised paper.
>
> &nbsp;
>
> **_3. "Using convolutional network to do dimension transferring from target X -> source X, using it for Orca-swin makes sense because they both belong to vision modeling, but why still use it for Orca-roberta?”_**
>
> We use convolutions for ORCA-RoBERTa because they are also effective for modeling long-range dependencies in 1D data, as shown by the following widely-used sequence models. For instance, the Temporal Convolutional Network [4] uses 1D convolutions to solve sequential MNIST, music modeling, and word-level language modeling problems, outperforming LSTMs and RNNs. The Structured State Spaces (S4) model [5] and SGConv [6] use 1D convolutions to achieve SOTA results on speech classification and the Long Range Arena benchmark [7]. Moreover, recent NAS work [8] has shown that equipping ResNets with various types of convolutions can result in highly effective models for all NAS-Bench-360 tasks (we compare ORCA with this method, DASH, in Table 2). Therefore, we use convolutional layers in the embedders for both 1D and 2D inputs (hence both RoBERTa and Swin) because: (1) they have been shown to be effective for feature extraction; (2) we want a unified way of defining the embedder architecture for diverse tasks, and using convolutions of different dimensions allows us to do so.
>
> &nbsp;
>
> **_4. "Why convolutional network is selected, and how does it compare to simple feed-forward network for dimension transformation, their comparison in down stream tasks, etc”_**
>
> Thanks for the suggestion. **Please see the third part of our general response for the embedder architecture ablations**, where we experimented with simple feed-forward layers and various convolutional architectures. We chose convolutional embedders for ORCA because they strike a balance between parameter-efficiency and downstream performance.

---

> > ### Author Response · Authors · 2022-11-12
> > **Response to Reviewer uNsA (cont'd)**
> >
> > **References**
> >
> > [1] Kumar et al., Fine-Tuning can Distort Pretrained Features and Underperform Out-of-Distribution, ICLR 2022.
> >
> > [2] Lee et al., Surgical Fine-Tuning Improves Adaptation to Distribution Shifts, 2022.
> >
> > [3] Rich Sutton, The Bitter Lesson, http://www.incompleteideas.net/IncIdeas/BitterLesson.html
> >
> > [4] Bai et al., An Empirical Evaluation of Generic Convolutional and Recurrent Networks for Sequence Modeling, 2018.
> >
> > [5] Gu et al., Efficiently Modeling Long Sequences with Structured State Spaces, ICLR 2022.
> >
> > [6] Li et al., What Makes Convolutional Models Great on Long Sequence Modeling? 2022.
> >
> > [7] Tay et al., Long Range Arena: A Benchmark for Efficient Transformers, 2020.
> >
> > [8] Shen et al., Efficient Architecture Search for Diverse Tasks, NeurIPS 2022.

---

### Author Response · Authors · 2022-11-12
**General Response: Novelty, Ablations, and Additional Baselines**

We appreciate the reviewers’ thoughtful feedback. In this comment, we first reiterate our main contribution, which is the first viable general-purpose cross-model transfer workflow that yields SOTA results on diverse tasks. Then, we present new ablation studies and baselines to show that our proposed first-align-then-fine-tune workflow, which has never been explored in the context of cross-modal learning, is the key to successful cross-modal transfer.

**1. ORCA Novelty**

Some reviewers expressed concerns about the novelty of individual components in our method, but we believe this overlooks the big picture novelty of our work, which is that **effective cross-modal transfer of pretrained transformers is only enabled by ORCA’s data alignment process through embedding learning. No previous work has studied such cross-modal alignment and generated such competitive models for diverse tasks as we have.** Several reviewers think our method might be simple. However, we believe that getting significantly better results than recently published works at top conferences is certainly novel and important. The fact that we used an intuitive and simple approach to achieve it only **increases** the impact. Below, we elaborate more on our contributions.
- **Practical problem setup:** ORCA is proposed as a broadly applicable cross-modal learning paradigm that can transfer language and vision transformers to non-text, non-vision tasks. None of the previous work has studied transfer learning in the context of diverse end tasks that span as many distinct domains, input dimensions, and output types as we do.
- **Workflow feasibility:** leveraging the key insight that data alignment enables effective cross-modal transfer, ORCA is the first approach that actually allows practitioners to obtain models better than hand-designed, AutoML-searched, and general-purpose architectures. Previous cross-modal methods either do not apply to the tasks we study or are ineffective as they do not address the structural and semantic differences between the end-task domain and the pretraining domain. Our distinctions relative to these works can be found in Table 1.
- **Insights for future research:** while we use an existing metric (OTDD) to do data alignment, we apply it in a new setting and generate new insights—notably, we observe that when the target domain differs significantly from the pretraining domain, naive fine-tuning may harm transfer, whereas aligning both data dimensionality and distribution before fine-tuning benefits transfer. We back up our use of OTDD with an ablation study that compares it with other metrics (see the next part of this comment).

**2. Ablation on Embedding Learning Metrics**

A common question from reviewers is why we chose OTDD as the embedding learning objective, so we present here an ablation study on the metrics that we have considered for minimizing distribution dissimilarity. The results show that (1) performing feature alignment generally helps downstream adaptation, regardless of which metric we minimize; (2) OTDD leads to the best overall performance, so we chose it for our workflow. Our findings confirm that it is the general idea of data alignment, rather than a specific metric, that makes cross-modal transfer work.

Specifically, we experiment with OTDD, maximum mean discrepancy (MMD) [1], the mutual-information-based TransRate [2], and Euclidean distance. We learn the embedders to minimize these metrics and then fine-tune the pretrained models. The test errors are as follows.

| |CIFAR-100 |Spherical |Darcy Flow |PSICOV |Cosmic |NinaPro |FSD50K|ECG |Satellite|DeepSEA|JSB Chorales| ListOps| Homology|
|:-:|:--:|:-:|:--:|:-:|:-:|:-:|:-:|:-:|:-:|:-:|:--:|:-:|:-:|
|OTDD          |**6.53**|**29.85**|**7.3E-3**|1.91|**0.21**|7.54|**0.56**|**0.29**|11.59|**0.29**|**2.44**|**51.9**|**87.21**|
|MMD           |6.62|33.64|7.4E-3|**1.9**|0.32|**7.48**|0.58|0.40|**11.29**|0.38|2.57|63.94|89.3|
|TransRate   |6.66|30.32|8E-3|1.92|0.31|8.64|0.57|0.44|11.35|0.38|2.56|63.02|97.21|
|Euclidean    |7.09|32.33|**7.3E-3**|1.91|0.27|7.51|0.59|0.41|11.4|0.34|2.56|60.2|89.97|
|No Embedder Learning |7.67|55.26|**7.3E-3**|1.92|0.24|8.35|0.63|0.44|13.86|0.51|2.59|72.79|89.31|

We see that OTDD has the lowest overall error rate, possibly because it considers *both* feature and label information, whereas the other metrics only consider the embedded feature distance. We also highlight that pretraining the embedder with any of the metrics can outperform standard fine-tuning without embedder pretraining on many tasks. This shows that bringing the target modality closer to the pretraining modality can aid cross-modal transfer.

---

> ### Author Response · Authors · 2022-11-12
> **General Response (cont'd)**
>
> **3. Ablation on Embedding Network Architectures**
>
> In addition to the embedding metric, we also investigate the effect of using different layers in the embedder, as suggested by reviewer uNsA. We experiment with a linear layer, a convolutional layer, and multiple convolutional layers. The results show that conv layers generally outperform linear layers, possibly because convolution is better at extracting features for a variety of data types, as shown by works that apply it to diverse tasks like PDE solving [3], genomic effect prediction [4], and time-series modeling [5]. To maximize both parameter efficiency and downstream efficacy, we chose a single conv layer for ORCA. (Note that there are other important operations in the embedder, such as pooling and flattening, to match the input dimensionality with the pretrained transformers, as discussed in Section 3.1.)
> | |ECG |Satellite|DeepSEA|JSB Chorales| ListOps| Homology|
> |:-:|:-:|:-:|:-:|:-:|:-:|:-:|
> |Linear                   |0.51|17.18|0.4|2.56|67.4|87.5|
> |1-Layer Conv        |**0.29** |11.59 |**0.29** |2.44 |**51.9**|**87.21**|
> |2-Layer Conv        |0.39|**11.58**|0.33|**2.43**|65.1|97.21|
>
> **4. Additional Baselines**
>
> Lastly, reviewers AGVC and tx7E wonder if we have a complete baseline comparison. To the best of our knowledge, we believe we have included the right and largest set of baselines spanning multiple categories such as hand-crafted models, AutoML methods, general-purpose models, and existing cross-modal transfer methods. Our results in Table 2 showed that ORCA achieves the best empirical performance.
>
> To complement existing evidence, we run additional experiments for (1) training RoBERTa and Swin from scratch using only the target data (as suggested by reviewer AGVC) and (2) prompting-based methods (as suggested by reviewer J4V2). The following results further validate ORCA’s efficacy.
>
> **4.1 Train-From-Scratch**
>
> | |CIFAR-100 |Spherical |Darcy Flow |PSICOV |Cosmic |NinaPro |FSD50K|ECG |Satellite|DeepSEA|JSB Chorales| ListOps| Homology|
> |:-:|:-:|:-:|:-:|:-:|:-:|:-:|:-:|:-:|:-:|:-:|:-:|:-:|:-:|
> |ORCA                      |**6.53**|**29.85**|**7.3E-3**|**1.91**|**0.21**|**7.54**|**0.56**|**0.29**|**11.59**|**0.29**|**2.44**|**51.9**|**87.21**|
> |Fine-tuning             |7.67|55.26|**7.3E-3**|1.92|0.24|8.35|0.63|0.44|13.86|0.51|2.59|72.79|89.31|
> |Train-from-scratch|50.87|76.67|8E-2|5.09|0.5|9.96|0.75|0.42|12.38|0.39|2.71|63.5|90.39|
>
> The fact that train-from-scratch is better than fine-tuning but worse than ORCA on tasks like Satellite, DeepSEA, and ListOps shows that when the target modality differs significantly from the pretraining modality, naive fine-tuning may harm transfer, but aligning the feature distribution using ORCA can resolve this issue and lead to positive transfer.
>
> **4.2 Prompting Methods**
>
> Most existing prompting methods are inapplicable to our tasks due to the cross-modal setting we are in and the difficulty of designing natural prompts for diverse data types. Reviewer J4V2 points us to a line of visual prompting work, so we experiment with two such algorithms, VP [6] and VPT [7]. They are only applicable to three tasks because the remaining ones either have inputs that cannot be reshaped to look like images or are just not classification tasks. We test VPT with the pretrained Swin-B Transformer (the same model we used for ORCA) and VP with the pretrained ResNet-50 (as the official implementation does not support vision transformers).
>
> | |Spherical|NinaPro|ECG|
> |:-:|:-:|:-:|:-:|
> |ORCA|**29.85**|**7.54**|**0.29**|
> |VP|98.05|33.18|0.57|
> |VPT |49.53|31.46|0.40|
>
> We see that visual prompting generally performs poorly on cross-modal tasks, possibly because the target modality differs significantly from the pretraining modality, in which case we must fine-tune the model weights to account for the modality difference.
>
> &nbsp;
>
> We will revise our paper to include the above results and the experiment details. Meanwhile, we welcome other baseline suggestions from the reviewers.
>
> &nbsp;
>
> **References**
>
> [1] Gretton et al., A Kernel Two-Sample Test, JMLR, 2012.
>
> [2] Huang et al., Frustratingly Easy Transferability Estimation, ICML 2022.
>
> [3] Li et al., Fourier Neural Operator for Parametric Partial Differential Equations, ICLR 2021.
>
> [4] Kelley et al., Sequential Regulatory Activity Prediction Across Chromosomes with Convolutional Neural Networks, Genome Research, 2018.
>
> [5] Li et al., What Makes Convolutional Models Great on Long Sequence Modeling? 2022.
>
> [6] Bahng et al., Exploring Visual Prompts for Adapting Large-Scale Models, 2022.
>
> [7] Jia et al., Visual Prompt Tuning, ECCV 2022.

---

### Author Response · Authors · 2022-11-16
**Rebuttal Revision Uploaded**

We appreciate the reviewers' constructive comments and have responded to all the questions and concerns. In addition, we have uploaded a revised version of our submission highlighting the changes made to address the comments, including:
- Abstract and introduction: emphasize our novelty in proposing a workflow that first enables effective cross-modal transfer via aligning the target data with the pretraining modality; improve presentation clarity
- Section 3.2: clarify embedder learning with OTDD
- Section 4.2: add ablation study for embedding learning metrics; add additional baselines

We are happy to address any new follow-up questions or concerns.

---

### Comment · Area_Chair_Q3hx · 2022-11-20
**Please update your reviews**

Please make sure that your reviews acknowledge authors’ responses and reflect your current evaluation of the paper. This is particularly important if you didn’t directly engage with the authors during the discussion phase (so the authors don’t know if their response changed your evaluation) or if you expressed an intention to update your rating but did not do so.

Cheers,
AC

---

### Decision · Program_Chairs · 2023-01-20

**Decision:**

Reject

**Justification For Why Not Higher Score:**

The novelty of proposed method cannot reach the bar of ICLR

**Justification For Why Not Lower Score:**

Good applications with extensive experiments.

**Metareview: Summary, Strengths And Weaknesses:**

Summary: This paper dives into the study of the pre-trained model (PM) finetuning and designs a cross-modal transfer method via minimizing OTDD between the source and target domain.

Strengths:
It should be acknowledged that how to adapt PM to diverse downstream tasks is a practical problem. The performance is promising.

Weakness: This paper still suffers from some deficiencies, as the reviewers pointed out. First, the methodology is mainly based on the existing OTDD, and no specific strategies are proposed (that is the biggest concern on its novelty. Simply applying the published method cannot have a great impact on the ICLR community.). Besides, the SOTA prompt-based baselines on PM finetuning are not included and compared, which cannot fully convince the effectiveness of the proposed method.

In summary, this paper is under the bar of ICLR and thus cannot be accepted in its current form.

**Summary Of Ac-Reviewer Meeting:**

I have called for a meeting for this paper discussion. However, Reviewer AGVC, uNsA and  WPBn are never responsive (I even tried to email them in separate emails for a 1-to-1 meeting, but no reply). So the meeting was held between AC, J4V2 and tx7E, and both of them are experts in this domain. This is borderline paper, so I also read through the paper. After the meeting, I also sent our meeting minutes of the rejection decision to all reviewers and solicited their opinions. Unfortunately, those reviewers who gave the high score haven't given any comments so far.

We all have a consensus that the novelty of the proposed method cannot reach the bar of ICLR (really nothing new, have little impact on the ICLR community). There are also missing some strong baselines to support the claims.